# TS$^2$: Training with Sparsemax+, Testing with Softmax for Accurate and Diverse LLM Fine-Tuning

**Ziyang Xu$^{1*}$, Ananthu Rajendran Pillai$^{2*}$, Yinghua Yao$^{34\dagger}$, Yuangang Pan$^{34}$**
$^1$National University of Singapore, Singapore
$^2$Nanyang Technological University, Singapore
$^3$Centre for Frontier AI Research, Agency for Science, Technology and Research, Singapore
$^4$Institute of High Performance Computing, Agency for Science, Technology and Research, Singapore
e1499203@u.nus.edu, ananthurpillai547@gmail.com,
{yao_yinghua,pan_yuangang}@a-star.edu.sg

## ABSTRACT

Large Language Models typically rely on Supervised Fine-Tuning (SFT) with Cross-Entropy (CE) loss to specialize in downstream tasks. However, CE forces the distribution toward one-hot targets and ignores alternative continuations, thereby limiting output diversity, a key drawback for generative applications that rely on sampling-based exploration. In this paper, we propose "Training with Sparsemax+, Testing with Softmax (TS$^2$)". Intuitively, sparsemax and its tailored loss mask the gradients of probabilities outside the support set, leaving excessive probability mass on irrelevant tail classes when evaluating with softmax. To address this issue, we propose an improved variant, Sparsemax+, for training, which augments the sparsemax loss with a suppression term that penalizes the out-of-support probabilities. At testing, we decode with softmax, yielding calibrated, non-degenerate probabilities where plausible near-ties survive. We fine-tuned Llama-3.1-8B and Qwen-2.5-7B with TS$^2$, achieving consistent improvements in accuracy and output diversity across chat, code, and open-domain benchmarks. Together, these results demonstrate that TS$^2$ provides a practical, drop-in solution for fine-tuning LLMs that are both more accurate and more creative. The code is available at https://github.com/xzy-bit/TS-2-ICLR-2026.

## 1 INTRODUCTION

Supervised fine-tuning (SFT) is one of the major steps in the Large Language Models (LLMs) post-training stage: with a small amount of high-quality annotated data, it teaches models to organize language better and produce instruction-following responses. The default loss function is cross-entropy loss, mainly because it coincides with maximum likelihood and is a strictly proper scoring rule, so minimizing it recovers the data generating conditional under well-specification (Gneiting & Raftery, 2007). However, the same geometry drives the probability mass toward the one-hot target and away from plausible alternatives, yielding overconfident posteriors and reduced useful diversity. A large body of work seeks to counteract this overconfidence and recover useful diversity. One branch changes only the decoding, e.g., nucleus sampling and best-of-$N$, leaving training dynamics and calibration untouched (Holtzman et al., 2020). Another branch alters the training signal itself. The recent GEM framework reframes SFT as reverse-KL minimization with an entropy regularizer, improving variety and mitigating overfitting (Li et al., 2025). These approaches highlight a fundamental issue: promoting diversity can conflict with keeping probabilities calibrated and tails disciplined.

We argue that the field lacks a precise operational notion of useful "diversity" for instruction following. In many tasks, we do not want to "spread probability" indiscriminately over the entire vocab-

---

$^*$Equal contribution. Work was done during their internship at Centre for Frontier AI Research, A*STAR.
$^\dagger$Corresponding author.

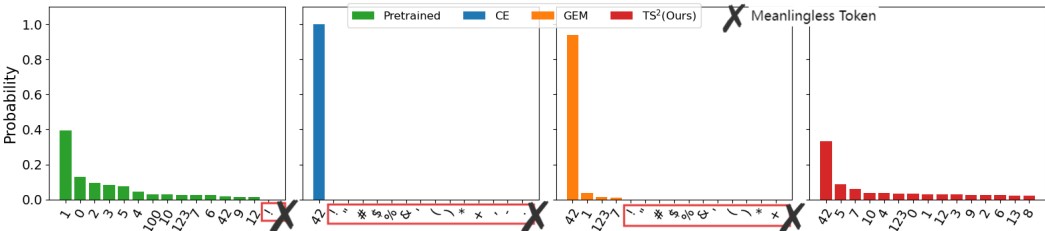

Figure 1: Token Distribution for single digit generation (detailed in Appendix C.4).

ulary. Instead, we want probability mass concentrated among a handful of semantically plausible next tokens, those with a real chance of leading to a high quality continuation, while aggressively deflating the long tail of obviously incorrect tokens toward (near) zero. The right diversity is *within the plausible set*, not across the whole simplex. The forward KL $\mathrm{KL}(p \,\|\, q)$ is mean-seeking, incentivizing probability wherever the data has support; the reverse KL $\mathrm{KL}(q \,\|\, p)$ is mode-seeking, concentrating mass on promising regions (Minka, 2005). This lens helps explain why CE with entropy maximization (a forward-KL-flavored objective under softmax) can inflate low-probability tokens, while reverse-KL flavored objectives like GEM avoid gratuitous tail mass. Yet even reverse-KL does not guarantee that clearly implausible tokens go to zero.

Our approach takes a geometric route by decoupling the mapping from logits to probabilities between training and testing. Specifically, we train with sparsemax and optimize a modified Fenchel–Young loss tailored to this mapping (Martins & Astudillo, 2016; Blondel et al., 2019), while at inference we revert to softmax, which restores calibrated and smooth probabilities on the same logits. Our tailored loss contains a tail penalty that drives non-support tokens to zero while ensuring the gold token is never penalized, even if it lies outside the instantaneous sparsemax support. Notably, CE with softmax collapses diversity: all non-gold logits, even plausible ones, are pushed toward zero. In contrast, sparsemax maintains a sparse support set by zeroing gradients of non-support tokens, preserving plausible candidates. However, if sparsemax were also used at inference, a converged model would still produce one-hot outputs, similar to CE with softmax decoding, thus limiting diversity (Martins & Astudillo, 2016; Blondel et al., 2019). Figure 1 shows that the pretrained model naturally exhibits diversity, but such diversity is lost during SFT: CE training drives the distribution into a one-hot solution, while GEM manages to retain only a few candidates, with most remaining probability mass assigned to irrelevant symbols. In contrast, our method delivers stronger and more stable diversity: the candidate set is both meaningful and varied, striking a balance between plausibility and coverage.

This decoupled recipe, Train with Sparsemax+, Test with Softmax (TS$^2$), has two key effects. During training, sparse gradients act as a principled early-stopping mechanism by avoiding wasted updates on already separated tail candidates. At inference, reverting to softmax restores smooth, calibrated probabilities so that plausible near-ties survive and sampling can explore them without aggressive temperature tuning. By construction, our method achieves local diversity among plausible tokens while assigning near-zero confidence to implausible ones. We position TS$^2$ among complementary strategies. Inference-only methods (e.g., nucleus, top-$k$, best-of-$N$) improve sample variety but leave training untouched; our approach reshapes training dynamics while remaining fully compatible with such decoders (Holtzman et al., 2020). Entropy targeting methods (e.g., GEM) promote spread but do not enforce exact zeros on implausible tokens; our penalty term supplies this "hard" suppression, while sparsemax ensures spread occurs where it matters (Li et al., 2025). Finally, because TS$^2$ decouples mappings rather than altering model architecture, it integrates seamlessly into existing SFT pipelines. The contribution of this paper are summarized in the following:

- We frame the problem as achieving Tail-Suppressed Plausible Diversity (TSPD) and propose TS$^2$, which decouples training and inference by using a Sparsemax+ loss with tail penalty for training and standard softmax for decoding.
- We provide a theoretical analysis showing how TS$^2$ avoids the distributional collapse common to CE training via a gradient-masking mechanism, thereby preserving diversity at inference.
- We demonstrate in practice that our TS$^2$ significantly improves winrates, sample efficiency in code generation, and output diversity across multiple benchmarks compared to existing methods.

## 2 DISTRIBUTION COLLAPSE AND OUR INSIGHT

Recent studies have observed an "alignment tax" in large language models (LLMs): while supervised fine-tuning (SFT) improves faithfulness and task adherence (Brown et al., 2020), it often comes at the cost of reduced output diversity and partial forgetting of pre-trained knowledge (O'Mahony et al., 2024; Kim et al., 2025). Pre-trained LLMs naturally exhibit a broad generative repertoire, producing multiple semantically valid outputs for the same prompt (Wang et al., 2025). However, after SFT, models tend to respond with highly deterministic and homogeneous output (Li et al., 2025), weakening their utility in downstream applications such as planning (Song et al., 2023), writing (Lee et al., 2022), or code generation (Liu et al., 2023), all of which fundamentally rely on the ability to explore diverse candidate responses.

A central obstacle in supervised fine-tuning is that cross-entropy (CE), driving the predictive distribution towards a one-hot distribution, causing all probability mass to collapse onto the gold token. This *distribution collapse* ensures convergence, but comes at a severe cost: the model suppresses all alternatives to nearly zero, leading to deterministic outputs, both in the choice of tokens and in the semantic content of the whole responses. The mechanism destroys *diversity*, erasing helpful variations preserved in the pre-trained distribution and thereby yielding monotonous generations.

### 2.1 OUR GUIDING INSIGHT: TAIL-SUPPRESSED PLAUSIBLE DIVERSITY

In generative modeling, output diversity is essential. A target distribution should retain a compact set of plausible candidates with non-negligible probability while suppressing irrelevant long-tail tokens toward zero. We formalize this as *Tail-Suppressed Plausible Diversity (TSPD)*, which remedies the distribution collapse commonly observed in existing SFT.

**Notation.** We consider prompt–response pairs $(x, y) \in \mathcal{D}$ from a supervised dataset $\mathcal{D}$. Let $f_\theta$ denote a pre-trained LLM parameterized by $\theta$. For a prompt $x$, let $\boldsymbol{z} = f_\theta(x) \in \mathbb{R}^K$ denote the corresponding logit vector[1]. We define the probability simplex as $\Delta^{K-1} = \{\boldsymbol{p} \in \mathbb{R}^K \mid p_i \geq 0, \sum_{i=1}^K p_i = 1\}$, where $\boldsymbol{p} = g(\boldsymbol{z})$ denotes a probability distribution obtained from the logits $\boldsymbol{z}$ via a probability mapping function $g(\cdot)$.

**Definition 1** (Tail-Suppressed Plausible Diversity $(m, \varepsilon_{\text{head}}, \varepsilon_{\text{tail}})$). *Given a prompt–response pair $(x, y)$, let $\boldsymbol{p} = g(f_\theta(x)) \in \Delta^{K-1}$ be a distribution over a vocabulary $\mathcal{V}$. Fix an integer $m \geq 2$ and thresholds $0 < \varepsilon_{\text{head}} \leq \frac{1}{m}$ and $0 \leq \varepsilon_{\text{tail}} \leq 1 - m \varepsilon_{\text{head}}$. Let $\text{Top}_m(\boldsymbol{p})$ denote the indices of the $m$ largest coordinates of $\boldsymbol{p}$. If $y \in \text{Top}_m(\boldsymbol{p})$, let $\mathcal{S} := \text{Top}_m(\boldsymbol{p})$; otherwise, let $\mathcal{S} := \text{Top}_{m-1}(\boldsymbol{p}) \cup \{y\}$. We say that $\boldsymbol{p}$ satisfies TSPD of order $m$ if*

$$\text{(Head Preservation)} \quad \min_{j \in \mathcal{S}} p_j \geq \varepsilon_{\text{head}}, \tag{1a}$$

$$\text{(Tail Suppression)} \quad \sum_{j \notin \mathcal{S}} p_j \leq \varepsilon_{\text{tail}}. \tag{1b}$$

*which ensures that candidates in $\mathcal{S}$ retain non-negligible probability, whereas tokens outside $\mathcal{S}$ receive essentially zero probability, thereby preserving uncertainty and transferable knowledge at inference. If one chooses $\varepsilon_{\text{head}} = 1/m$ exactly, then the strict requirement $\varepsilon_{\text{tail}} \geq 0$ forces $m \varepsilon_{\text{head}} = 1$ and $p_j = 0 \, \forall j \notin \mathcal{S}$; therefore, in practice one can take $\varepsilon_{\text{head}} < 1/m$ and relax $\varepsilon_{\text{tail}} > 0$.*

**Corollary 1.** *If Definition 1 holds and $\varepsilon_{\text{tail}} < \varepsilon_{\text{head}}$, then $\max_{j \notin \mathcal{S}} p_j \leq \varepsilon_{\text{tail}} < \varepsilon_{\text{head}} \leq \min_{i \in \mathcal{S}} p_i$, so each plausible sample has strictly higher probability than any tail sample.*

**Corollary 2.** *If all probability mass collapses onto the ground-truth token, i.e., $p_y = 1$ and $p_{y'} = 0 \, \forall y' \neq y$, then $\boldsymbol{p}$ fails to qualify the TSPD $(m(\geq 2), \varepsilon_{\text{head}}, \varepsilon_{\text{tail}})$.*

In the next section, we motivate our method that operationalizes this principle, directly countering the diversity-reducing bias of CE loss while retaining the benefits of supervised fine-tuning.

---

[1]$y$ and $x$ can be sequential, where an auto-regressive formulation is used.

# 3  ACHIEVING TAIL-SUPPRESSED PLAUSIBLE DIVERSITY

A natural way to realize TSPD in Equation (1) is to exploit the sparsity of the sparsemax mapping sparsemax($\boldsymbol{z}$) (Martins & Astudillo, 2016), which projects logits $\boldsymbol{z} \in \mathbb{R}^K$ onto the probability simplex $\Delta^{K-1}$ and can assign exact zeros to non–support tokens.

Let $\boldsymbol{z} \in \mathbb{R}^K$ and let $z_{(1)} \geq z_{(2)} \geq \cdots \geq z_{(K)}$ denote the sorted coordinates of $\boldsymbol{z}$. Define $k(\boldsymbol{z}) :=$ $\max \left\{ k \in \{1, \ldots, K\} \middle| 1 + k\, z_{(k)} > \sum_{j=1}^{k} z_{(j)} \right\}$, and the threshold $\tau(\boldsymbol{z}) = \frac{\sum_{j=1}^{k(\boldsymbol{z})} z_{(j)} - 1}{k(\boldsymbol{z})}$.

The sparsemax probabilities are then given elementwise by

$$p_i^{\mathsf{sp}}(\boldsymbol{z}) = \mathrm{sparsemax}(\boldsymbol{z})_i := \max\{z_i - \tau(\boldsymbol{z}), 0\}, \quad i = 1, \ldots, K.$$

The (data-dependent) support set is $S^{\mathsf{sp}}(\boldsymbol{z}) = \left\{ i \in \{1, \ldots, K\} : p_i^{\mathsf{sp}}(\boldsymbol{z}) > 0 \right\}$, which follows from the definition above and no circularity arises. Equivalently, in vector form, $\boldsymbol{p}^{\mathsf{sp}}(\boldsymbol{z}) = \left[ \boldsymbol{z} - \tau(\boldsymbol{z})\mathbf{1} \right]_+$, where $[\boldsymbol{v}]_+ := \max\{\boldsymbol{v}, 0\}$ is applied elementwise.

In effect, sparsemax automatically identifies a compact support set of plausible candidates $S^{\mathsf{sp}}(\boldsymbol{z})$ and prunes away the long tail. Compared to the softmax probability mapping $\boldsymbol{p}^{\mathsf{sf}}(\boldsymbol{z}) = \frac{\exp(\boldsymbol{z})}{\sum_{i=1}^{K} \exp(z_i)} := \mathrm{softmax}(\boldsymbol{z})$, its Jacobian is sparse; in particular, gradients vanish outside $S^{\mathsf{sp}}(\boldsymbol{z})$ when the target lies in the support (Lemma 3).

**Lemma 3** (Gradients vanish outside the sparsemax support (Martins & Astudillo, 2016)). *Let* $\boldsymbol{p} = \mathrm{sparsemax}(\boldsymbol{z})$ *and* $S^{\mathsf{sp}}(\boldsymbol{z})$ *be its support. Define* $\mathcal{L}(\boldsymbol{p}, y)$ *as a supervised loss between the sparsemax probability* $\boldsymbol{p}$ *and the target* $y$. *If* $y \in S^{\mathsf{sp}}(\boldsymbol{z})$, *then* $\forall i \notin S^{\mathsf{sp}}(\boldsymbol{z})$, $\frac{\partial \mathcal{L}(\boldsymbol{z}, y)}{\partial z_i} = 0$.

While sparsemax provides margin-induced sparsity, it nonetheless tends to collapse into a nearly one-hot distribution once the leading logit surpasses the margin threshold. Such collapse inevitably reduces sampling diversity, making sparsemax undesirable for inference.

This motivates us to instead carry out decoding with softmax. Under this choice, the gradient-vanishing property established in Lemma 3 remains advantageous during training: by nullifying gradients outside the active support whenever the target is included, it mitigates the cross-entropy–style erosion of plausible near-optimal alternatives, thereby inducing an implicit early-stopping effect.

**Theorem 4** (Sparsemax expands pairwise gaps faster than softmax). *Let* $\boldsymbol{z} \in \mathbb{R}^K$, $\boldsymbol{p}^{\mathsf{sf}} = \mathrm{softmax}(\boldsymbol{z})$, *and* $\boldsymbol{p}^{\mathsf{sp}} = \mathrm{sparsemax}(\boldsymbol{z})$. *For any indices* $i \neq j$, *let* $u := z_i - z_j$ *and we have*

$$\frac{\partial}{\partial u}(p_i^{\mathsf{sp}} - p_j^{\mathsf{sp}}) = 1 \quad \forall i, j \in \mathcal{S}^{\mathsf{sp}}, \qquad \text{sparsemax}$$

$$\frac{\partial}{\partial u}\left(p_i^{\mathsf{sf}} - p_j^{\mathsf{sf}}\right) < 1, \qquad \text{softmax}$$

Given the same logits, Theorem 4 shows that sparsemax linearly preserves pairwise probability gaps within its active support and collapses to a one-hot prediction once a finite margin is attained, whereas softmax strictly contracts such gaps. Consequently, sparsemax induces sharp discrimination and faster label collapse during training, while applying softmax to the same logits at inference preserves non-degenerate mass on plausible candidates, maintaining output diversity that is desirable for generative tasks.

**Corollary 5** (Softmax remains TSPD-valid when sparsemax is one-hot). *Let* $\boldsymbol{z} \in \mathbb{R}^K$ *with* $y = \arg\max_j z_j$, *and* $\delta_j := z_y - z_j$. *Assume sparsemax is one-hot at* $y$, *i.e.,* $\delta_{\min} := \min_{j \neq y} \delta_j \geq \gamma > 0$ *(e.g.,* $\gamma = 1$*), and the top-$m$ head is bounded:* $\delta_{(k)} := z_y - z_{(k)} \leq B \ \forall k = 2, \ldots, m$. *Set* $A_m = m + (K - m)e^{-\gamma}$. *Then for* $p^{\mathsf{sf}} = \mathrm{softmax}(\boldsymbol{z})$ *we have*

$$p_y^{\mathsf{sf}} \geq \frac{1}{A_m}, \quad p_{(k)}^{\mathsf{sf}} \geq \frac{e^{-B}}{A_m} \ (\forall k = 2, \ldots, m), \quad \sum_{k > m} p_{(k)}^{\mathsf{sf}} \leq \frac{(K - m)e^{-\gamma}}{A_m}.$$

*Consequently,* $p^{\mathsf{sf}}$ *satisfies TSPD of order* $m$ *with any thresholds* $0 < \varepsilon_{\mathrm{head}} \leq \frac{e^{-B}}{A_m}$, $\frac{(K-m)e^{-\gamma}}{A_m} \leq \varepsilon_{\mathrm{tail}} < 1 - m\,\varepsilon_{\mathrm{head}}$.

**Remark 1.** *Without the head bound $\delta_{(k)} \leq B$ ($\forall k \leq m$), $p^{\mathsf{sf}}_{(m)}$ can be made arbitrarily small even when $\delta_{\min} \geq 1$, so only a vanishingly small head floor $\varepsilon_{\mathrm{head}}$ can be guaranteed for general $m$.*

**Remark 2** (Existence of a tight upper bound for the cumulated tail mass $\sum_{k>m} p^{\mathsf{sf}}_{(k)}$ in Corollary 5)**.** *Under the assumption of Corollary 5, let $\Omega_{\min} = 1 + (m-1)e^{-B} + (K-m)e^{-\gamma}$, then we have $\sum_{k>m} p^{\mathsf{sf}}_{(k)} \leq \frac{(K-m)e^{-\gamma}}{\Omega_{\min}}$. With the new upper bound, $p^{\mathsf{sf}}$ still satisfies TSPD of order $m$ with any thresholds $0 < \varepsilon_{\mathrm{head}} \leq \frac{e^{-B}}{A_m}$, $\frac{(K-m)e^{-\gamma}}{\Omega_{\min}} \leq \varepsilon_{\mathrm{tail}} \leq 1 - m\,\varepsilon_{\mathrm{head}}$. (see Appendix D for detailed derivation.).*

According to Remark 2, the upper bound on the cumulated tail mass $\sum_{k>m} p^{\mathsf{sf}}_{(k)}$ is strictly increasing in $K$ and approaches 1 as $K \to \infty$. Importantly, this bound is tight (see Proposition 6 in Appendix D): there exists a worst-case configuration where the tail mass grows monotonically with $K$. Thus, for large vocabularies, the admissible tail under softmax at inference becomes nearly 1, indicating that sparsemax training has not theoretically guaranteed the suppression of irrelevant tail mass, contradicting the goal of suppressing irrelevant tail mass.

To address these issues, we propose a fine-tuning strategy of **Training with Sparsemax+, Testing with Softmax**. Sparsemax+ builds on Sparsemax, inheriting margin-induced sparsity to introduce gradient masking during training, thereby implicitly enforcing an early-stopping effect once the top-1 candidate is clearly separated. It further incorporates a lightweight *Tail-suppressing Loss* to explicitly penalize residual probability on tail tokens, ensuring that tail mass is sharply suppressed. At inference, we revert to softmax over the same logits, which restores smooth, calibrated probabilities across the plausible candidates within the support set, while keeping the irrelevant tail mass negligible due to the additional suppressing effect. In this way, the model learns to *separate and prune* the logits during training, yet *preserve and diversify* the output distribution during inference, achieving the desired support-aware diversity.

## 4 TS$^2$: TRAINING WITH SPARSEMAX+, TESTING WITH SOFTMAX

In the following, we present supervised fine-tuning based on the Fenchel-Young loss, which encompasses both the softmax and sparsemax mappings. It then motivates our Sparsemax+ loss.

### 4.1 DIFFERENT PREDICTION MAPPINGS WITH THE UNIFIED FENCHEL-YOUNG LOSS

For any strictly convex regularization function $\Pi : \Delta^{K-1} \to \mathbb{R}$, the corresponding regularized prediction function is $\boldsymbol{p}_*(\boldsymbol{z}) = \arg\max_{\boldsymbol{p} \in \Delta^{K-1}} \langle \boldsymbol{p}, \boldsymbol{z} \rangle - \Pi(\boldsymbol{p})$. The associated Fenchel-Young loss can be represented as

$$L_\Pi(\boldsymbol{z}; y) = \Pi(\boldsymbol{e}_y) - \Pi(\boldsymbol{p}_*) + \langle \boldsymbol{z}, \boldsymbol{p}_* - \boldsymbol{e}_y \rangle, \tag{3}$$

where $y$ is the gold label and $\boldsymbol{e}_y$ is the corresponding one-hot vector. Different choices of $\Pi$ yield different prediction mappings and losses.

**Softmax** Softmax corresponds to using the negative Shannon entropy as regularizer $\Pi(\boldsymbol{p}) = \sum_{i=1}^{K} p_i \log p_i$, which gives $\boldsymbol{p}_*(\boldsymbol{z}) = \frac{\exp(\boldsymbol{z})}{\sum_{i=1}^{K} \exp(z_i)} := \mathrm{softmax}(\boldsymbol{z})$. The Fenchel-Young loss reduces to the standard CE loss $L_{\mathrm{softmax}}(\boldsymbol{z}; y) = \log \sum_{i=1}^{K} \exp(z_i) - z_y = -\log \frac{\exp(z_y)}{\sum_{i=1}^{K} \exp(z_i)}$.

**Sparsemax** Sparsemax corresponds to using the negative Gini entropy as regularizer $\Pi(\boldsymbol{p}) = \frac{1}{2} \sum_{i=1}^{K} p_i(1 - p_i)$, which gives $\boldsymbol{p}_*(\boldsymbol{z}) = [\boldsymbol{z} - \tau(\boldsymbol{z})\boldsymbol{1}]_+ := \mathrm{sparsemax}(\boldsymbol{z})$. The corresponding Fenchel-Young loss, called the sparsemax loss, is $L_{\mathrm{sparsemax}}(\boldsymbol{z}; y) = -z_y + \frac{1}{2} \sum_{j \in S^{\mathsf{sp}}(\boldsymbol{z})} \left( z_j^2 - \tau^2(\boldsymbol{z}) \right) + \frac{1}{2}$.

In conclusion, when training with sparsemax but performing inference with softmax, although $\mathrm{softmax}(\boldsymbol{z})$ does not yield a one-hot output like $\mathrm{sparsemax}(\boldsymbol{z})$, it still assigns the highest probability to the correct class. Importantly, it naturally enables early stopping and preserves distributional diversity across all classes, which is consistent with the goal of diversifying plausible candidates.

Given prompt–response pairs $(x, y)$ from a supervised dataset, let $\boldsymbol{z} \in \mathbb{R}^K$ be a logit vector and $\boldsymbol{p}$ be probability mapping either via softmax or sparsemax. If the gradient of the sparsemax loss

vanishes, i.e., $\nabla_{\boldsymbol{z}} \mathcal{L}_{\text{sparsemax}}(\boldsymbol{z}; y) = 0$, then it follows that $\text{sparsemax}(\boldsymbol{z})_y = 1$. For any index $\forall j \neq y$, $\text{sparsemax}(\boldsymbol{z})_j = 0$, it holds that $\text{softmax}(\boldsymbol{z})_j > 0$. That is, softmax assigns non-zero probability to all entries, including those which sparsemax maps to zero. According to Corollary 5, the cumulated tail mass of softmax outside the top-$m$ satisfies $\sum_{k>m} p_{(k)}^{\text{sf}} \leq \frac{(K-m)e^{-\gamma}}{A_m}$. With large vocabularies, the admissible tail under softmax at inference becomes nearly 1. This behavior is undesirable, as assigning non-negligible probabilities to clearly incorrect classes may lead the model to produce semantically meaningless outputs.

**Sparsemax+** To address this issue, we introduce a lightweight tail-suppressing loss that explicitly suppresses probabilities assigned to the non-plausible candidates. Given logits $\boldsymbol{z} \in \mathbb{R}^K$, let $\boldsymbol{p}^{\text{sf}} = \text{softmax}(\boldsymbol{z}) \in \Delta^{K-1}$. The tail-suppressing loss is defined as

$$\mathcal{L}_{\text{sup}}(\boldsymbol{p}; y) = -\log(1 - \sum_{i \notin \mathcal{S}} p_i^{\text{sf}}),$$

where $\mathcal{S}$ is defined in Definition 1. This penalty drives the probabilities of tail tokens toward zero, thereby avoiding residual mass on clearly implausible candidates.

**Remark 3.** *The tail suppressing loss can be interpreted as a direct generalization of the standard softmax CE to the* group-label *setting. Specifically, given logits $\boldsymbol{z}$ and softmax distribution $\boldsymbol{p}^{\text{sf}} = \text{softmax}(\boldsymbol{z})$, the suppressing term can be written as*

$$L_{\text{sup}}(\boldsymbol{z}) = -\log(1 - \sum_{i \notin S} p_i^{\text{sf}}) = -\log \sum_{i \in S} p_i^{\text{sf}},$$

*which is exactly the softmax cross-entropy with the target label being the merged "super-class" $S$. In the special case where $S = \{y\}$ is a singleton, this reduces to the usual CE loss $-\log p_y^{\text{sf}}$. Thus, the suppressing loss can be viewed as encouraging the softmax probability mass to concentrate on a set of plausible candidates while retaining the probabilistic interpretation of cross-entropy.*

Combining sparsemax with the tail-suppressing loss yields our proposed *Sparsemax+ loss*:

$$L_{\text{spm+}}(\boldsymbol{z}; y) = -z_y + \frac{1}{2} \sum_{j \in S^{\text{sp}}(\boldsymbol{z})} \left(z_j^2 - \tau^2(\boldsymbol{z})\right) + \alpha \left(-\log\left(1 - \sum_{i \notin S^{\text{sp}}(\boldsymbol{z}), i \neq y} p_i^{\text{sf}}\right)\right), \quad (4)$$

where $\tau(\boldsymbol{z})$ is the sparsemax threshold and $\alpha > 0$ controls the strength of the suppression. For simplicity, we find that directly implementing the candidate set $\mathcal{S}$ from Definition 1 using the sparsemax support $S^{\text{sp}}(\boldsymbol{z})$ achieves superior performance.

We summarize our fine-tuning strategy of **Training with Sparsemax+, Testing with Softmax** in Algorithm 1. From $L_{\text{spm+}}(\boldsymbol{z}; \boldsymbol{y})$ in equation 4, we see that it prevents CE-style erosion of plausible near-ties by amplifying relative ratios among top logits while nulling the rest, thereby achieving two goals: sparsemax selects a stable support set with early stopping of gradient flow, and the suppressing term explicitly drives unreasonable tokens toward zero to prevent spurious mass at inference.

## 5 EXPERIMENTS

To situate our work within the current state-of-the-art, we build upon the experimental foundation of GEM (Li et al., 2025), adopting a similar training setup. Our primary methodological difference is the substitution of the GEM objective with our proposed TS$^2$ loss. Furthermore, while GEM evaluates OpenLLM Leaderboard tasks using a standard one shot setting, we employ a multi-response, best-of-N protocol. We argue this is a more faithful and informative evaluation for diversity aware models, as it measures model's latent ability to find the correct answer rather than penalizing it for plausible "hesitation" in a single attempt.

**Setup.** We conduct experiments on two powerful, open source base models: Llama-3.1-8B and Qwen-2-7B . For supervised finetuning, we use the high quality `UltraFeedback` dataset (Cui et al., 2024), a large-scale corpus of preference aligned responses generated by a diverse set of models. All models are finetuned for 3 epochs using the AdamW optimizer with an effective batch size of 128. We employ a cosine learning rate schedule with an initial rate of $2 \times 10^{-5}$ and a warm-up ratio of 0.03, a standard practice for fine-tuning modern LLMs (Yu et al., 2024; Liu et al.,

---

**Algorithm 1** $\text{TS}^2$: Training with Sparsemax+, Testing with Softmax

---

    **Input:** pre-trained model $f_\theta$; training dataset $\mathcal{D}_{tr} = \{(x, y)\}$; test dataset $\mathcal{D}_{te} = \{x\}$.
    **Hyperparameters:** epochs $T$; batch size $B$; learning rate $\eta > 0$; suppression weight $\alpha > 0$.
1: **for** $t = 1$ to $T$ **do**                                                                 ▷ Training Phase
2:     **for** mini-batch $\{(\boldsymbol{x}_b, \boldsymbol{y}_b)\}_{b=1}^{B} \subset \mathcal{D}_{tr}$ **do**
3:         Compute logits $\boldsymbol{z}_b \leftarrow f_\theta(\boldsymbol{x}_b), \forall b = 1, 2, \ldots, B$
4:         Compute loss $L_b \leftarrow L_{\text{spm+}}(\boldsymbol{z}_b; \boldsymbol{y}_b), \forall b = 1, 2, \ldots, B$            ▷ Sparsemax+ loss
5:     **end for**
6:     Update $\theta \leftarrow \theta - \eta \nabla_\theta \frac{1}{B} \sum_{b=1}^{B} L_b$
7: **end for**
8: **for** test input $\boldsymbol{x} \in \mathcal{D}_{te}$ **do**                                                ▷ Testing Phase
9:     Compute logits $\boldsymbol{z} \leftarrow f_\theta(\boldsymbol{x})$
10:    Predict probability $\boldsymbol{p} \leftarrow \text{softmax}(\boldsymbol{z})$
11:    Evaluation on $\boldsymbol{p}$                                                 ▷ Use for decoding
12: **end for**

---

2024). The maximum sequence length is capped at 2,048 tokens. For our proposed $\text{TS}^2$ method, the suppression weight $\alpha$ (see Equation 4) is empirically determined for each model architecture, with optimal values reported alongside results. Further implementation details are provided in the Appendix B.

We compare $\text{TS}^2$ against a suite of strong and relevant baselines to provide a comprehensive evaluation: **Cross-entropy (CE):** The standard SFT objective, which serves as our primary baseline. **CE with Weight Decay (CE+WD):** A common regularization technique shown to help preserve diversity in instruction tuning (Ouyang et al., 2022; Bai et al., 2022). We use a weight decay coefficient of 0.1. **NEFTune (NEFT):** A regularization method that adds noise to word embeddings during training to mitigate overfitting and improve generalization (Jain et al., 2023). **GEM:** The current state-of-the-art method for diversity preserving SFT, which we use as our main point of comparison (Li et al., 2025).

## 5.1 IMPROVING ACCURACY AND DIVERSITY IN OPEN-ENDED GENERATION

We first evaluate $\text{TS}^2$ in open ended domains to assess its ability to navigate the critical trade-off between response quality and diversity. While standard fine-tuning often improves quality at the cost of collapsing the output distribution, we hypothesize that $\text{TS}^2$ can break this trade-off by simultaneously enhancing generation quality and fostering a rich, useful diversity beneficial for sampling-based decoding. To test this, we evaluate on two distinct benchmarks. For conversational chat, we use the `AlpacaEval` dataset (Dubois et al., 2024) with a best-of-32 (BoN@32) protocol; a state-of-the-art reward model, `FsfairX-LLaMA3-RM-v0.1` (Lambert et al., 2024), selects the best response, which is then compared against GPT-4 to determine a win rate. For code generation, we measure the pass@k metric on the `HumanEval` benchmark (Chen et al., 2021), which assesses the model's ability to generate functionally correct Python code via execution.

| Model | Method | Win Rate (%) ↑ | N-gram ↑ | 100 - Self-BLEU ↑ | Sent-BERT ↑ |
|---|---|---|---|---|---|
| | CE | 29.77 | 17.78 | 47.04 | 9.97 |
| | CE+WD | 29.72 | 17.78 | 47.14 | 10.03 |
| LLaMA-3.1-8B | NEFT | 29.77 | 17.74 | 47.41 | 10.07 |
| | GEM | 31.53 | 20.32 | 49.82 | 11.16 |
| | **TS$^2$ (Ours)** | **33.12** | **23.78** | **53.87** | **12.80** |
| | CE | 31.41 | 17.23 | 16.77 | 7.95 |
| | CE+WD | 31.05 | 17.43 | 17.08 | 8.06 |
| Qwen-2-7B | NEFT | 30.36 | 16.59 | 24.59 | 8.06 |
| | GEM | 33.89 | 24.35 | 31.19 | 9.25 |
| | **TS$^2$ (Ours)** | **37.48** | **30.15** | **39.04** | **9.81** |

Table 1: Win rate (Best of N@32) and diversity metrics for Llama-3.1-8B and Qwen-2-7B on AlpacaEval. $\text{TS}^2$ achieves the best results across both quality and diversity on both architectures.

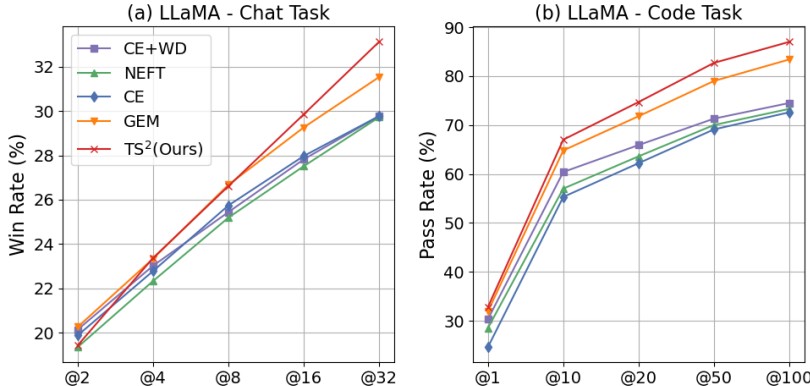

Figure 2: Performance of Llama-3.1-8B on open-ended tasks. Left: Win rate on AlpacaEval vs. sampling budget (N). Right: Pass rate on HumanEval vs. sampling budget (k). $TS^2$ consistently outperforms baselines.

**Performance on Chat and Code Generation.** As shown in Figure 2, $TS^2$ demonstrates a clear performance advantage on Llama-3.1-8B. In chat generation, its win rate at a budget of N=32 responses, reaches 33.12%, which is an improvement of 11.2% relative over the baseline cross entropy loss and a 5.0% relative improvement over the strong GEM baseline. This advantage extends to structured problem solving, on `HumanEval`, $TS^2$ achieves a pass@100 of 87.00%, which is 4.3% increase relative to GEM and 19.8% to that of CE. Notably, the diversity fostered by our method translates to superior sample efficiency: the pass@50 rate for $TS^2$ (82.70%) nearly matches GEM's pass@100 performance (83.40%), indicating that correct solutions can be found with fewer samples. Similar results are also observed for Qwen-2-7B model. Detailed breakdown of results for both Llama-3.1-8B and Qwen-2-7B are detailed in the Table 8.

Crucially, these performance gains do not come at the cost of diversity. As detailed in Table 1, $TS^2$ not only achieves the highest win rate but also scores best across all three diversity metrics. It improves N-gram diversity by 17.0%, BLEU diversity by 8.1% and sentence-bert diversity by 10.7% over GEM for LLama-3.1-8B. Similarly for Qwen-2-7B, the same metrics are improved by 23.8%, 25.1% and 6% respecitevely over GEM. This result confirms that $TS^2$ successfully breaks the quality-diversity trade-off, producing responses that are simultaneously judged as higher quality by a reward model while being measurably more varied.

### 5.1.1 DIVERSITY ON CREATIVE WRITING TASKS

To further probe the nature of the diversity generated by $TS^2$, we evaluate it on purely creative tasks: generating poems from 573 titles in the `poetry8` dataset and stories from 500 prompts from `ROCStories` (Mostafazadeh et al., 2016). As shown in Table 2, $TS^2$ once again achieves the highest scores across all three diversity metrics on both tasks, confirming its ability to produce a wider range of high-quality, creative outputs compared to all baselines.

| Method | Poem | | | Story | | |
|---|---|---|---|---|---|---|
| | N-gram ↑ | 100 - Self-BLEU ↑ | Sent-BERT ↑ | N-gram ↑ | 100 - Self-BLEU ↑ | Sent-BERT ↑ |
| CE | 38.87 | 55.38 | 14.83 | 44.47 | 67.20 | 22.15 |
| CE+WD | 38.92 | 55.69 | 14.17 | 44.43 | 67.26 | 22.22 |
| NEFT | 38.80 | 55.68 | 14.13 | 44.31 | 67.21 | 22.04 |
| GEM | 46.59 | 57.50 | 14.70 | 50.05 | 69.15 | 24.02 |
| $TS^2$ (Ours) | **49.70** | **59.41** | **16.52** | **52.10** | **70.36** | **24.98** |

Table 2: Diversity evaluation on creative writing tasks for Llama-3.1-8B. Higher is better.

### 5.2 PRESERVING PRE-TRAINED CAPABILITIES ON STANDARD BENCHMARKS

To assess generalization and knowledge retention, we evaluate models on six tasks from the Open-LLM Leaderboard: ARC, GSM8K, HellaSwag, MMLU, TruthfulQA, and WinoGrande. Instead of the standard greedy one-shot decoding that penalizes models preserving multiple reasoning paths,

we propose a best-of-n (BoN) strategy on the OpenLLM leaderboard, which is better aligned with evaluating the capabilities of diversity-preserving models.

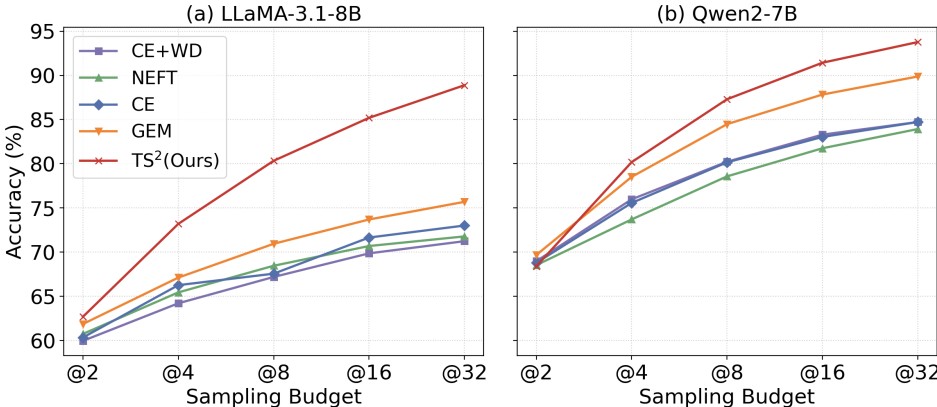

Figure 3: Average Best-of-N accuracy across six OpenLLM Leaderboard tasks. While competitive in few-shot settings (@2), TS$^2$'s performance scales far more effectively with the sampling budget, revealing its superior knowledge retention.

We argue that a more faithful metric is Best-of-N (BoN) accuracy. This protocol measures the model's latent ability to identify the correct answer within a small sampling budget, which better reflects the true underlying capabilities of a well-calibrated, diverse model. For fair comparision, all methods are evaluated under the same BoN protocol and we report the average accuracy across all tasks.

Figure 3 validates this hypothesis. While all methods are competitive at a small sampling budget, TS$^2$'s performance scales significantly better as 'N'(responses) increases. On Llama-3.1-8B, the average accuracy of TS$^2$ at N=32 reaches 88.88%, a massive 13.2-point absolute (+17.4% relative) improvement over GEM (75.69%). The trend is consistent on Qwen-2-7B, where TS$^2$ again achieves the highest accuracy, demonstrating the robustness of our TS$^2$ across different model architectures.

This shows that TS$^2$ effectively preserves the model's pre-trained knowledge. Unlike CE, which collapses the distribution and discards valid alternatives, TS$^2$ maintains a clean, calibrated set of high-quality reasoning paths. With sampled responses, the model consistently finds the correct solution. A detailed breakdown of performance on each of the six tasks is provided in the Table 10.

### 5.3 ABLATION STUDY

To assess the contribution of each component, we run an ablation study on AlpacaEval, comparing win rate against GPT-4 and BLEU diversity. TS$^2$ integrates three elements: (1) sparsemax-based training, (2) softmax decoding, and (3) a tail-suppression penalty. We evaluate three variants: **Decoupling Only** (sparsemax training, softmax inference, no penalty), **Unified Sparsemax** (sparsemax for both training and inference), and **Suppression Only** (CE loss with suppression term).

Figure 4 demonstrates that all components of TS$^2$ are essential. First, using the Decoupling Only strategy results in a massive increase in diversity, high BLEU diversity score, but a catastrophic drop in win rate. This shows that while decoupling unlocks variety, the suppression penalty is crucial for ensuring that this diversity is high-quality and not just uncalibrated noise.

Conversely, the Unified Sparsemax approach achieves a competitive win rate but offers lesser diversity than our full method. This confirms that the switch to softmax at inference is key to translating the learned logit geometry into a rich, sample-able probability distribution. Finally, applying the Suppression Only penalty to a standard CE baseline fails on both metrics, proving it is not a standalone improvement but works in synergy with the sparsemax-defined support set.

Meanwhile, the TS$^2$ method successfully integrates these components, achieving the best balance of high win rate and high diversity across both model architectures. This analysis confirms that the sparsemax objective, the decoupled inference, and the suppression penalty are all necessary and synergistic elements of our approach.

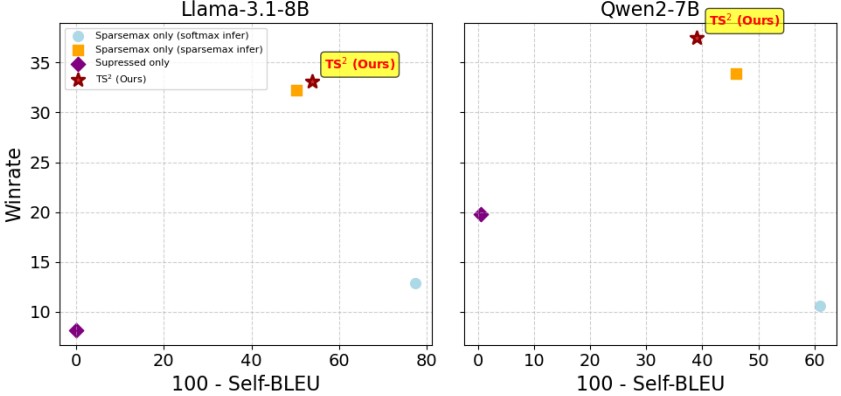

Figure 4: Ablation study on Llama-3.1-8B and Qwen-2-7B.

## 6 CONCLUSION

In this work, we make the first step toward decoupling training and inference by adopting different prediction mappings in supervised finetuning. By combining **Sparsemax+ loss**; a tailored design that leverages margin induced sparsity with an additional suppression term for non plausible tokens; with softmax decoding at inference, our approach achieves significant improvements over existing SFT paradigms. It preserves support-aware diversity while maintaining high accuracy, thereby alleviating the alignment tax. Despite its simplicity, our method consistently outperforms CE and GEM across both chat and code tasks, achieving the highest win rates and more diverse generations. Unlike prior methods that inevitably trade off diversity against accuracy, our paradigm improves both, providing a natural remedy to distribution collapse and open up new directions for advancing alignment with broad and long term impact.

## ETHICS STATEMENT

This work investigates new algorithms for supervised fine-tuning of large language models. Our objective is to improve training stability and output diversity, thereby broadening the range of downstream applications. The methods introduced in this paper are purely algorithmic and evaluated on public datasets.

## REPRODUCIBILITY STATEMENT

Experiment details for reproducing our numerical results can be found in Appendix B and Appendix C.

## LLM USAGE STATEMENT

We used large language model to correct grammar errors, polish the writing, and adjust the formatting of the paper.

## ACKNOWLEDGMENT

This work was supported by the National Research Foundation, Singapore under its National Large Language Models Funding Initiative (AISG Award No: AISG-NMLP-2024-004). Any opinions, findings and conclusions or recommendations expressed in this material are those of the author(s) and do not reflect the views of National Research Foundation, Singapore.

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

## A    RELATED WORK

Our work, TS$^2$, intersects with three primary areas of research: supervised finetuning (SFT) and its inherent limitations, methods for enhancing generative diversity in large language models (LLMs), and the use of sparse activation functions in neural networks.

### A.1    SUPERVISED FINETUNING AND THE ALIGNMENT TAX

Supervised finetuning is a major landmark in adapting pre-trained LLMs to downstream applications, enabling them to follow instructions and adhere to specific conversational styles (Ouyang et al., 2022; Touvron et al., 2023). The standard practice involves minimizing a cross-entropy (CE) loss on a dataset of high quality datasets. While effective, this approach is known to induce an "alignment tax" (O'Mahony et al., 2024), where models become overly specialized to the finetuning distribution. This often leads to a reduction in creative capacity, a phenomenon sometimes termed "knowledge forgetting" or a collapse in output diversity (Kim et al., 2025; Li et al., 2025). The CE objective, by driving the model's posterior towards a one-hot representation of the target token, aggressively penalizes all alternative continuations, including those that are semantically plausible. This results in overconfident and deterministic models. Our work directly addresses this limitation by replacing the CE objective with a loss that preserves a set of plausible next-tokens, thereby mitigating the distributional collapse and retaining more of the pre-trained model's capabilities.

Overconfidence and miscalibration have been extensively studied in deep neural networks more broadly, where modern architectures are known to produce poorly calibrated probability estimates even when their accuracy is high (Guo et al., 2017). Recent work has begun to document and address similar phenomena in large language models and conversational agents. Mielke et al. (2022) show that dialogue agents can be systematically overconfident and propose linguistic calibration strategies to temper their stated certainty, Kadavath et al. (2022) analyze when LLMs "know what they know" and how their confidence estimates relate to factual correctness, and Tian et al. (2023) and Lin et al. (2022) develop post-hoc calibration methods for RLHF and SFT-trained LLMs by verbalized confidence or calibration probes. These approaches primarily operate on top of a fixed model, either

by rescaling scalar confidences or modifying prompts at inference time. By contrast, TS$^2$ targets the source of overconfidence in SFT itself: we reshape the token-level logit geometry during training so that plausible alternatives are preserved within a compact support set while the softmax tail is explicitly suppressed.

## A.2 ENHANCING GENERATIVE DIVERSITY

Efforts to counteract the loss of diversity in finetuned LLMs can be broadly categorized into **decoding-time** and **training-time** strategies.

**Decoding-Time Strategies:** A popular line of work focuses on modifying the sampling process at inference. Techniques such as **temperature scaling**, **top-k sampling**, and **nucleus (top-p) sampling** (Holtzman et al., 2020) manipulate the output probability distribution to encourage variety. Similarly, **best-of-N sampling**, where multiple candidate responses are generated and ranked by a reward model (Bai et al., 2022), can improve output quality by exploring a wider search space. While widely used and effective, these methods are applied post-hoc and do not address the underlying overconfidence of the model's learned distribution. TS$^2$ is complementary to these techniques but fundamentally different, as it reshapes the logit geometry during training to produce a more inherently diverse and well-calibrated posterior.

**Training-Time Strategies:** Another branch of research modifies the training objective itself. **Label smoothing** (Szegedy et al., 2016) is a regularization technique that discourages overconfidence by training on soft targets. More recently, unlikelihood training was proposed to explicitly penalize undesirable tokens or repetitive patterns (Welleck et al., 2019). Closest to our work is the recent **GEM framework** (Li et al., 2025), which recasts SFT as a reverse-KL minimization problem with an entropy regularizer. GEM successfully improves diversity by preventing the model's posterior from collapsing. However, it does not enforce a hard separation between plausible and implausible tokens, potentially leaving residual mass on the long tail of the distribution. TS$^2$ offers a more direct approach: the sparsemax function provides a principled mechanism for identifying a compact support set of plausible tokens, while our proposed suppression penalty explicitly drives the probability of out-of-support tokens to zero, achieving a cleaner and more decisive separation.

## A.3 SPARSE ACTIVATIONS IN NEURAL NETWORKS

The sparsemax function, which we leverage for our training objective, is a projection onto the probability simplex that can produce exact zeros (Martins & Astudillo, 2016). It was originally introduced as a sparse alternative to softmax for attention mechanisms and structured prediction tasks, valued for its ability to select a small subset of relevant inputs. The sparsemax loss is a specific instance of a Fenchel-Young loss, a broader class of losses that provides a unified framework for various structured prediction mappings (Blondel et al., 2019). While sparsemax has been explored for classification and attention, its application to generative LLM fine-tuning for diversity preservation is novel. Critically, our work is the first to propose a decoupled paradigm: we use the desirable properties of sparsemax (e.g., gradient masking for non-support tokens) during training but revert to the smooth, fully-supported softmax for inference. This decoupling is the key to unlocking calibrated diversity, a concept not explored in prior work that typically uses the same mapping for both training and testing.

## B EXPERIMENT DETAILS

We conduct all training on H200-141GB GPUs, employing the DeepSpeed framework with ZeRO-2 optimization and gradient checkpointing enabled. Offloading is disabled. For efficient and reproducible training, we adopt flash-attention-2 with deterministic backward passes. Our base models are `Llama-3.1-8B` and `Qwen-2-7B`, optimized using AdamW with a total batch size of 128. The learning rate is initialized at $2 \times 10^{-5}$ with a warm-up ratio of 0.03 and follows a cosine decay schedule, as suggested by prior work (Yu et al., 2024; Liu et al., 2024; Cui et al., 2024),. Training is run for 3 epochs. All supervised datasets are reformatted into the chat style with the `Llama-3.1-8B-Instruct` and `Qwen2.5-7B-Instruct` tokenizer. For inference, we employ `vLLM` to accelerate response generation.

The supervised finetuning is done on the binarized `UltraFeedback` dataset curated by the HuggingfaceH4 team[2], which contains 61,135 training examples and 1,000 held-out test prompts. Inputs longer than 2,048 tokens are truncated, while shorter ones are padded. To achieve a global batch size of 128, we use 4 GPUs, each with a per-device batch size of 8 and gradient accumulation of 4. A single training run requires roughly 12 GPU hours. For CE+WD baselines, the weight decay coefficients is 0.1. For NEFT, we set the noise scale to 5, consistent with Jain et al. (2023).

**Evaluation Protocol** For chatting, we use 805 prompts from `AlpacaEval` and score outputs with the `FsfairX-LLaMA3-RM-v0.1` reward model. The maximum decoding length is 2,048, and each prompt yields 32 samples using temperature=0.6, top-$k$=50, and top-$p$=0.9. Win rate is computed against GPT-4[3] responses via the Bradley–Terry model:

$$P(y \succ y'|x) = \frac{\exp(r(x,y))}{\exp(r(x,y)) + \exp(r(x,y'))}.$$

For code generation, we adopt the `HumanEval` benchmark (164 Python problems). Prompts follow the template of (Wei et al., 2024).

```
You are an exceptionally intelligent coding assistant that consistently
delivers accurate and reliable responses to user instructions.
@@ Instruction
{instruction}
```

For each task, we sample 200 outputs with the same decoding configuration. The evaluation metric is pass rates, which are computed using execution-based evaluation scripts from Magicoder[4].

## C  ADDITIONAL RESULTS

### C.1  SENSITIVITY OF HYPER-PARAMETERS

Our TS$^2$ framework provides two hyper-parameters to control the final behavior:

- **Training-time control.** The hyperparameter $\alpha$ in Sparsemax+ equation 4 determines the strength of suppression during training.
- **Training time control via logit scale $c$.** There is also an internal logit scale $c$ in $L_{\text{spm+}}(c\boldsymbol{z}; y)$, which plays a role analogous to temperature. In all reported experiments we simply set $c = 1$, which already yields strong performance.
- **Inference-time control.** The inference of our TS$^2$ relies on softmax. Therefore, the temperature parameter can also be adjusted to influence the final output. We fixed it to 1 which works well.

In our formulation, the hyperparameter $\alpha$ directly controls how aggressively the Sparsemax+ transformation suppresses negative (low-logit) components. When $\alpha \to 0$, there's no supersession, the loss is the pure sparsemax loss, lots of low logits keep their mass, yielding a extreme diverse output(see Fig.6). As $\alpha$ increases, the suppression let distribution becomes more concentrated on the top few logits, which will let the diversity fall.The results are showing in Tab.3. In practice, we typically choose $0 < \alpha < 1$, which allows the first part dominates when updating.

### C.2  EXTRA BASELINES

For completeness, we additionally evaluate two overconfidence-mitigation baselines that are commonly used as alternatives to standard cross-entropy: CE + label smoothing(0.2) and $\alpha$-Entmax ($\alpha$ = 1.5). Both methods modify the training distribution to reduce overconfident predictions. Tab.4 reports the AlpacaEval (Chat) win-rate(BoN@32) results across all methods on Llama-3.1-8b.

---

[2]`https://huggingface.co/datasets/HuggingFaceH4/ultrafeedback_binarized`
[3]`https://github.com/tatsu-lab/alpaca_eval/tree/main/results/gpt4_1106_preview`
[4]`https://github.com/ise-uiuc/magicoder/blob/main/experiments/text2code.py`

Table 3: Effect of $\alpha$ on Chat ($c$=1.0, Llama-3.1-8b).

| Setting | Winrate (%) | n-gram | 100-self-BLEU | Sent-BERT |
|---------|-------------|--------|---------------|-----------|
| $\alpha$=0.25 | **33.12** | **23.78** | **53.87** | **12.80** |
| $\alpha$=0.50 | 31.44 | 20.54 | 51.34 | 11.61 |
| $\alpha$=0.75 | 30.84 | 19.07 | 49.29 | 10.96 |
| $\alpha$=1.00 | 30.39 | 18.13 | 48.05 | 10.30 |
| $\alpha$=5.00 | 25.34 | 13.44 | 41.42 | 8.12 |

Table 4: Comparison of TS$^2$, GEM, CE, sparsemax, Entmax, and CE with label smoothing on chat task(Alpaca-Eval).

| | TS$^2$ (ours) | GEM | CE | CE + label smoothing | sparsemax | 1.5-Entmax |
|---|---------------|-----|-----|----------------------|-----------|------------|
| Winrate (%) | **33.12** | 31.53 | 29.77 | 28.25 | 12.87 | 13.51 |

While both CE + label smoothing and 1.5-Entmax introduce additional diversity compared to vanilla CE, this comes at a substantial loss in win-rate, and both remain far below GEM and TS$^2$. Sparsemax and Entmax also do not explicitly regulate cumulative tail probability: label smoothing over-flattens the distribution, whereas sparsemax/entmax produce sparse supports without suppressing softmax-style tails at inference. These behaviors reduce helpfulness and alignment quality.

In contrast, TS$^2$ combines a sparse training objective with an explicit tail-suppression mechanism while retaining softmax decoding. This yields the highest win-rate among all compared methods, while still improving diversity, suggesting that TS$^2$ achieves a more favorable balance between confidence calibration, output variability, and instruction-following performance.

## C.3 MULTI-TURN TASK PERFORMANCE

We additionally evaluate our method on MT-Bench-101, a fine-grained multi-turn dialogue benchmark covering detailed conversational capabilities[5]. After generating the response of each question, we using `GPT-4.1 nano` to score them from 13 metrics, see Table 5. Across all capability groups, our method achieves consistent improvements, for example, +0.95 in CM, +1.67 in CC, and +1.18 in SC—demonstrating gains in reasoning coherence, contextual consistency, and conversational stability. Complete results are shown in Table 6. These findings indicate that our approach generalizes effectively to multi-turn dialogue scenarios.

## C.4 MACRO- AND MICRO-LEVEL ANALYSIS OF TOKEN DISTRIBUTIONS

To understand why our method simultaneously improves accuracy and diversity, we analyze token probability distributions from two complementary perspectives: (i) a *macro-level* analysis of model outputs on a real benchmark, and (ii) a *micro-level* controlled probing task.

**Macro-level distribution.** We evaluate the models(`Llama`) on the `AlpacaEval` dataset. For each generated response, we record the probability of every selected token and compute the average probability of that response. We then plot these values across all responses to obtain a global view of the distribution. As shown in Figure 5, CE exhibits the highest mean probability ($\approx 0.90$) with the smallest variance, indicating collapsed and overly uniform predictions. GEM lowers the mean probability to about 0.86 with a larger variance, consistent with its entropy-regularized updates that discourage overconfidence. Moving along the sequence CE $\rightarrow$ GEM $\rightarrow$ Sparsemax (sparse inference) $\rightarrow$ TS$^2$, we observe a systematic trend: mean probability decreases (remaining above 0.8), while variance increases, revealing a more balanced allocation of probability mass to plausible alternatives.

---

[5]https://github.com/mtbench101/mt-bench-101

| Abbr. | Task | Description |
|---|---|---|
| CM | Context Memory | Recall early dialogue details to address the user's current query. |
| SI | Separate Input | First turn provides task requirements; later turns supply the actual input. |
| AR | Anaphora Resolution | Identify pronoun referents across dialogue turns. |
| TS | Topic Shift | Detect and follow the new topic when users change subjects unexpectedly. |
| CC | Content Confusion | Avoid interference from similar-looking but semantically distinct queries. |
| CR | Content Rephrasing | Rephrase the content of the previous response based on new user requirements. |
| FR | Format Rephrasing | Reformat the previous response into the structure requested by the user. |
| SC | Self-Correction | Correct the previous response according to user feedback. |
| SA | Self-Affirmation | Retain the previous response when the user provides inaccurate feedback. |
| MR | Mathematical Reasoning | Solve multi-turn mathematical reasoning tasks collaboratively. |
| GR | General Reasoning | Solve multi-turn general reasoning tasks collaboratively. |
| IC | Instruction Clarification | Ask clarifying questions when the user's query is under-specified. |
| PI | Proactive Interaction | Proactively ask questions to sustain or deepen the conversation. |

Table 5: Descriptions of the 13 MT-Bench-101 dialogue capability dimensions.

| Model | Avg. | Perceptivity | | | | | Adaptability | | | | | | Interactivity | |
|---|---|---|---|---|---|---|---|---|---|---|---|---|---|---|
| | | Memory | Understanding | | Interference | | Rephrasing | | Reflection | | Reasoning | | Questioning | |
| | | CM | SI | AR | TS | CC | CR | FR | SC | SA | MR | GR | IC | PI |
| CE | 5.99 | 5.01 | 4.59 | 6.03 | 5.10 | 5.03 | 7.33 | 7.02 | 6.34 | 7.38 | 6.37 | 4.67 | 7.49 | 5.46 |
| GEM | 6.24 | 4.63 | 4.43 | 6.88 | 5.95 | 5.36 | 7.19 | 7.76 | 7.2 | 8.05 | 6.09 | 5.05 | 6.54 | 5.98 |
| TS$^2$(Ours) | **6.65** | **5.96** | **4.76** | 6.58 | 5.94 | **6.70** | **8.27** | 6.60 | **7.42** | **8.53** | 6.03 | **6.14** | 6.80 | **6.76** |

Table 6: Comparison on MT-Bench-101.

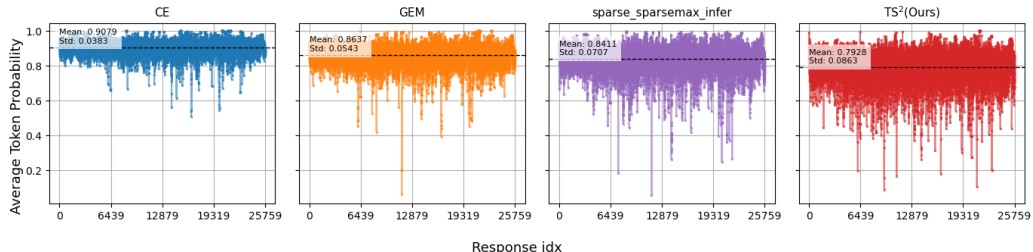

Figure 5: Macro-level analysis: average selected token probability distribution on `AlpacaEval`.

**Micro-level probing.** To complement the macro-level view, we design a controlled probing task to test whether models can distribute probability mass across relevant candidates. We prompt the model with the few-shot instruction to generate a single-digit number. Each model is queried 100 times. Whenever a digit is generated, we record the probability distribution of the top-300 tokens. Finally, we compute the average probability of each token across the 100 trials, resulting in a fine-grained view of how probability mass is allocated.

```
You're an AI assistant, I will give you an example of following question.
Example:
User: Give me a word of fruit.
Assistant: Apple.
Now you follow the format of the example,
Give me a single-digit number,
Answer:
```

The results, shown in Figure 1, reveal stark differences. **CE** collapses to a one-hot distribution: the chosen digit monopolizes probability, while the tail is filled with irrelevant tokens. **GEM** retains a few candidate digits but remains nearly one-hot, yielding limited diversity. **Sparsemax (Sparsemax-infer)** distributes mass across more digits, but still assigns non-negligible probability to spurious tokens. In contrast, **TS**$^2$ combines sparsemax, which preserves probability on relevant

digits, with the suppressing loss, which eliminates unrelated characters. This synergy results in distributions that are both diverse and accurate.

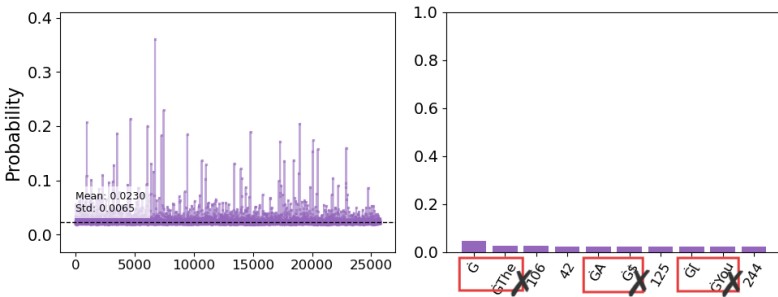

Figure 6: Macro- and Micro-level probling: Sparsemax Training and Softmax Inference

As a special case, we also examine the strategy of **sparsemax training with softmax inference**(shown in Figure 6). In the *macro-level probing*, this setting produces a distribution that is close to uniform, suggesting that the model does not exhibit clear preferences over candidate tokens. In the *micro-level probing task*, we observe that although some valid numerical answers (such as "42" or "125") appear, a large number of irrelevant tokens also receive comparable probability mass. As a result, the model's outputs become difficult to interpret, and its effective generation ability is diminished. This illustrates why conventional diversity metrics may report artificially high scores in this case: while probability is spread across many tokens, much of it corresponds to spurious rather than meaningful outputs.

## C.5 HARD THRESHOLD

| CE | 0.25-top3 | 0.5-top3 | 1.0-top3 | 0.5-top5 | 0.5-top10 | 1.0-top10 | 0.25-TS$^2$ | 0.5-TS$^2$ |
|---|---|---|---|---|---|---|---|---|
| 50.00 | 50.16 | 52.58 | 48.80 | 49.86 | 47.58 | 46.61 | **53.73** | 51.03 |

Table 7: Results on `Llama-3.2-1B` with different $\alpha$ and sparsification strategies. All tokens except target are thrown out the support set.

We also evaluate the `Llama-3.2-1B` under different values of $\alpha$ and supersession strategies. Here, "top-$k$" means only the largest $k$ logits are preserved in the defined support set, while all other tokens (except the target) are thrown out the support set, and "TS$^2$" denotes our proposed two-stage suppression method. We take the vanilla cross-entropy (CE) training as the baseline, which yields a score of 50.

From the results Tab. 7, we observe two main trends: (i) Increasing $\alpha$ from 0.25 to 1.0 generally decreases performance, indicating that larger $\alpha$ values reduce the model's output diversity. (ii) Within the top-$k$ setting, smaller $k$ (e.g., top-3 vs. top-10) leads to higher diversity and better scores, while larger $k$ values dilute the distribution and hurt performance. Overall, both reducing $\alpha$ and carefully selecting smaller $k$ encourage the model to maintain useful diversity, while our TS$^2$ method further boosts results beyond simple top-$k$ truncation.

## C.6 CHAT AND CODE GENERATION

Table 8 details the performance of both models on the open-ended generation tasks. For chat generation, it presents the win rate against GPT-4 across various best-of-N sampling budgets (N = 2, 4, 8, 16, 32). For code generation, it shows the corresponding pass@k rates for k = 1, 10, 20, 50, and 100.

## C.7 CREATIVE WRITING

We further investigate output diversity on two creative writing tasks: poetry and short stories. For poetry, we use 573 titles drawn from the Huggingface `poetry8` dataset, which covers themes such

| Chat | LLaMA-3.1-8B | | | | | Qwen-2-7B | | | | |
|------|------|------|------|------|------|------|------|------|------|------|
| | CE+WD | NEFT | CE | GEM | TS$^2$($\alpha = 0.25$) | CE+WD | NEFT | CE | GEM | TS$^2$($\alpha = 0.5$) |
| @2 | 20.14 | 19.35 | 19.88 | **20.26** | 19.43 | 18.35 | 18.13 | 18.4 | 18.06 | **18.72** |
| @4 | 23.02 | 22.33 | 22.78 | 23.34 | **23.37** | 21.59 | 21.49 | 21.78 | 21.93 | **22.54** |
| @8 | 25.44 | 25.19 | 25.74 | **26.67** | 26.61 | 24.58 | 24.39 | 27.9 | 26.26 | **27.66** |
| @16 | 27.82 | 27.51 | 27.97 | 29.26 | **29.85** | 27.76 | 27.02 | 27.9 | 30.02 | **32.77** |
| @32 | 29.77 | 29.72 | 29.77 | 31.53 | **33.12** | 31.05 | 30.36 | 31.41 | 33.89 | **37.48** |

| Code | LLaMA-3.1-8B | | | | | Qwen-2-7B | | | | |
|------|------|------|------|------|------|------|------|------|------|------|
| | CE+WD | NEFT | CE | GEM | TS$^2$($\alpha = 0.25$) | CE+WD | NEFT | CE | GEM | TS$^2$($\alpha = 0.5$) |
| @1 | 30.30 | 28.50 | 24.60 | 31.90 | **32.80** | 45.10 | 45.30 | **44.90** | 41.80 | 42.20 |
| @10 | 60.40 | 57.00 | 55.30 | 64.80 | **67.00** | 76.80 | 76.50 | 76.00 | **78.50** | 78.20 |
| @20 | 65.90 | 63.60 | 62.20 | 71.80 | **74.70** | 81.30 | 81.00 | 81.10 | 84.50 | **84.60** |
| @50 | 71.30 | 70.00 | 69.10 | 79.00 | **82.70** | 84.10 | 83.20 | 83.50 | 87.20 | **87.80** |
| @100 | 74.50 | 73.30 | 72.60 | 83.40 | **87.00** | 87.20 | 85.40 | 86.60 | 90.20 | **91.50** |

Table 8: Performance comparison of different methods on LLaMA-3.1-8B and Qwen-2-7B models for the chat and code geneartion tasks

as love, nature, and mythology. For stories, we construct 500 prompts from the ROCStories dataset (Mostafazadeh et al., 2016). In both settings, the instruction is to write a piece titled "[X]" in under 200 words, where [X] is sampled from the corresponding dataset.

Diversity is measured along three dimensions following Kirk et al. (2024): **N-gram**, the fraction of distinct n-grams within a single response (intra-diversity); **Self-BLEU**, computed by treating each sample as the reference for the others (inter-diversity); **Sentence-BERT dissimilarity**, the mean cosine distance between generated responses in the embedding space. All scores are scaled to the range [0, 100], with higher values indicating greater diversity.

For evaluation, each model generates 16 completions per prompt using temperature=0.6, top-$k$=50, and top-$p$=0.9. Results are summarized in Table 9. It is evident that methods such as CE+WD and NEFT bring only marginal improvements in diversity. GEM consistently improves intra- and inter-diversity, while TS$^2$ achieves the highest scores.

| Method (Llama-3.1-8B) | Poem | | | Story | | |
|------|------|------|------|------|------|------|
| | N-gram ↑ | 100 - Self-BLEU ↑ | Sent-BERT ↑ | N-gram ↑ | 100 - Self-BLEU ↑ | Sent-BERT ↑ |
| CE+WD | 38.92 | 55.69 | 14.17 | 44.43 | 67.26 | 22.22 |
| NEFT | 38.80 | 55.68 | 14.13 | 44.31 | 67.21 | 22.04 |
| CE | 38.87 | 55.38 | 14.83 | 44.47 | 67.20 | 22.15 |
| GEM | 46.59 | 57.50 | 14.70 | 50.05 | 69.15 | 24.02 |
| TS$^2$ ($\alpha = 0.25$) | **49.70** | **59.41** | **16.52** | **52.10** | **70.36** | **24.98** |

| Method (Qwen-2-7B) | Poem | | | Story | | |
|------|------|------|------|------|------|------|
| | N-gram ↑ | 100 - Self-BLEU ↑ | Sent-BERT ↑ | N-gram ↑ | 100 - Self-BLEU ↑ | Sent-BERT ↑ |
| CE+WD | 44.29 | 44.9 | 8.66 | 56.62 | 50.06 | 19.01 |
| NEFT | 44.37 | 45.09 | 8.55 | 59.66 | 52.2 | 18.94 |
| CE | 43.94 | 44.92 | 8.56 | 56.44 | 49.83 | 18.86 |
| GEM | 50.29 | 48.62 | 9.54 | 60.91 | 56.05 | 20.98 |
| TS$^2$ ($\alpha = 0.5$) | **53.46** | **51.10** | **10.26** | **62.13** | **57.17** | **20.95** |

Table 9: Diversity evaluation on creative writing tasks (poem and story). Higher values indicate greater diversity (N-gram, 100 - Self-BLEU, and Sentence-BERT.

## C.8 OpenLLM Leaderboard Tasks

Table 10 reports results on six representative OpenLLM leaderboard tasks under varying sampling budgets. These benchmarks collectively reflect a broad spectrum of model capabilities: ARC focuses on grade-school science questions, reflecting *commonsense reasoning*; GSM8K requires multi-step solutions, capturing *mathematical reasoning*; HellaSwag emphasizes physical commonsense and narrative continuation, probing *contextual understanding*; MMLU spans 57 subjects, testing *broad*

*factual knowledge*; `TruthfulQA` challenges models with common misconceptions, measuring *robustness*; and `WinoGrande` is a coreference benchmark, assessing *pronoun disambiguation and fine-grained language understanding*.

Building on this setup, we observe that other methods exhibit only limited gains as the sampling budget increases. In contrast, TS$^2$ consistently improves performance across tasks, achieving the largest boosts under larger budgets, often surpassing all baselines by a substantial margin. The improvements are especially pronounced for `LLaMA-3.1-8B`, where diversity-oriented training translates into 10–15 point gains under BoN sampling. For `Qwen-2-7B`, whose baseline win rates already exceed 90%, the relative gains appear smaller but still confirm the benefits of preserving diversity during training.

(a) Llama-3.1-8B

| Method | ARC@2 | ARC@4 | ARC@8 | ARC@16 | ARC@32 | Hella@2 | Hella@4 | Hella@8 | Hella@16 | Hella@32 |
|---|---|---|---|---|---|---|---|---|---|---|
| CE+WD | 75.59 | 80.27 | 80.27 | 83.28 | 83.61 | 66.95 | 70.55 | 72.95 | 74.37 | 75.05 |
| NEFT | 75.67 | 79.26 | 81.27 | 81.61 | 81.61 | 65.45 | 69.53 | 72.17 | 73.66 | 74.35 |
| CE | 76.59 | 79.50 | 71.27 | 82.60 | 83.60 | **67.03** | 61.96 | 63.06 | 63.85 | 64.33 |
| GEM | 78.60 | 82.27 | 83.94 | 85.28 | 85.61 | 66.51 | 71.71 | 74.84 | 76.96 | 77.95 |
| TS$^2$($\alpha = 0.25$) | **78.93** | **85.95** | **88.96** | **90.30** | **91.63** | 65.47 | **76.76** | **84.43** | **88.55** | **90.81** |

| Method | Wino@2 | Wino@4 | Wino@8 | Wino@16 | Wino@32 | MMLU@2 | MMLU@4 | MMLU@8 | MMLU@16 | MMLU@32 |
|---|---|---|---|---|---|---|---|---|---|---|
| CE+WD | 59.65 | 61.48 | 63.30 | 64.01 | 64.96 | 60.75 | 63.12 | 65.42 | 66.93 | 67.98 |
| NEFT | 61.24 | 63.14 | 64.64 | 66.46 | 66.77 | 61.20 | 63.85 | 66.24 | 68.03 | 69.19 |
| CE | 59.75 | **80.27** | 80.27 | 83.28 | 83.61 | 60.86 | 63.98 | 66.23 | 67.90 | 68.90 |
| GEM | 61.80 | 64.64 | 66.69 | 68.43 | 69.85 | 62.04 | 66.12 | 69.32 | 71.85 | 73.54 |
| TS$^2$($\alpha = 0.25$) | **66.14** | 75.77 | **80.51** | **83.98** | **87.21** | **64.19** | **73.64** | **81.24** | **85.82** | **88.42** |

| Method | Truth@2 | Truth@4 | Truth@8 | Truth@16 | Truth@32 | GSM@2 | GSM@4 | GSM@8 | GSM@16 | GSM@32 |
|---|---|---|---|---|---|---|---|---|---|---|
| CE+WD | 43.02 | 45.29 | 46.88 | 48.59 | 49.20 | 53.84 | 64.59 | 74.37 | 82.03 | 86.66 |
| NEFT | 46.74 | 50.06 | 51.41 | 52.39 | 53.12 | 54.21 | **66.87** | 75.06 | 82.03 | 85.67 |
| CE | 43.21 | 45.04 | 47.86 | 49.82 | 50.67 | **54.66** | **66.87** | **76.72** | 82.49 | 86.96 |
| GEM | 47.86 | 51.29 | 55.32 | 57.04 | 59.73 | 54.44 | 66.72 | 75.59 | **82.64** | 87.49 |
| TS$^2$($\alpha = 0.25$) | **51.16** | **62.42** | **71.85** | **80.05** | **87.39** | 50.27 | 64.67 | 75.06 | 82.49 | **87.87** |

(b) Qwen2-7B

| Method | ARC@2 | ARC@4 | ARC@8 | ARC@16 | ARC@32 | Hella@2 | Hella@4 | Hella@8 | Hella@16 | Hella@32 |
|---|---|---|---|---|---|---|---|---|---|---|
| CE+WD | 84.61 | 87.62 | 88.96 | 89.96 | 89.96 | 80.20 | 85.50 | 88.76 | 90.67 | 91.44 |
| NEFT | **84.94** | 87.29 | 88.62 | 89.96 | 89.96 | 80.20 | 85.39 | 88.67 | 90.59 | 91.40 |
| CE | 84.61 | 87.95 | 89.62 | 90.30 | 90.30 | **80.45** | 85.60 | 88.79 | 90.62 | 91.42 |
| GEM | **84.94** | 88.96 | 92.30 | 92.60 | 93.64 | 80.01 | **87.54** | 91.94 | 94.52 | 95.62 |
| TS$^2$($\alpha = 0.5$) | 83.27 | **89.63** | **92.97** | **93.97** | **94.31** | 75.39 | 87.36 | **94.18** | **96.97** | **98.21** |

| Method | Wino@2 | Wino@4 | Wino@8 | Wino@16 | Wino@32 | MMLU@2 | MMLU@4 | MMLU@8 | MMLU@16 | MMLU@32 |
|---|---|---|---|---|---|---|---|---|---|---|
| CE+WD | 70.79 | 77.34 | 81.84 | 83.34 | 83.89 | 69.42 | 77.86 | 84.53 | 90.37 | 93.56 |
| NEFT | 71.19 | 77.26 | 82.00 | 83.58 | 83.97 | 69.28 | 71.49 | 79.83 | 85.97 | 91.28 |
| CE | 70.63 | 77.26 | 82.16 | 83.66 | 84.13 | 69.37 | 76.65 | 83.89 | 89.46 | 93.40 |
| GEM | **73.63** | 81.76 | 87.05 | 89.50 | 90.37 | **69.91** | 80.36 | 88.86 | 93.86 | 96.82 |
| TS$^2$($\alpha = 0.5$) | 71.11 | **84.37** | **91.31** | **95.26** | **95.97** | 69.01 | **82.11** | **90.67** | **94.84** | **97.65** |

| Method | Truth@2 | Truth@4 | Truth@8 | Truth@16 | Truth@32 | GSM@2 | GSM@4 | GSM@8 | GSM@16 | GSM@32 |
|---|---|---|---|---|---|---|---|---|---|---|
| CE+WD | 45.65 | 49.69 | 52.75 | 55.07 | 55.93 | 66.72 | 77.86 | 84.53 | 90.37 | 93.56 |
| NEFT | 45.41 | 49.32 | 52.50 | 54.46 | 55.69 | 67.32 | 71.49 | 79.83 | 85.97 | 91.28 |
| CE | 45.16 | 49.32 | 52.63 | 54.71 | 55.69 | 68.16 | 76.65 | 83.89 | 89.46 | 93.40 |
| GEM | **46.87** | 52.14 | 57.89 | 62.66 | 65.97 | 68.08 | 80.36 | 88.86 | 93.86 | 96.82 |
| TS$^2$($\alpha = 0.5$) | 46.63 | **55.55** | **64.01** | **72.70** | **78.82** | **68.99** | **82.11** | **90.67** | **94.84** | **97.65** |

Table 10: Pass Rate (%) of Different Methods on 6 OpenLLM leaderboard tasks under Various Sampling Budgets.

## C.9 DIVERSE OUTPUT EXAMPLES

**Chatting Examples.** To qualitatively illustrate the behavioral differences between CE-SFT and TS$^2$-SFT, we randomly sampled 32 generations from each model on AlpacaEval for the prompt **"I want to get better at networking at work"**. From these sets, we selected representative examples to showcase typical response patterns for both training methods. The results are shown in Tab. 11

| Prompt | I want to get better at networking at work. |
|---|---|
| **CE-SFT** | **Resp. 1: Networking is an** essential skill for personal and professional growth. **Here are some tips to help you improve your networking skills at work: (1) identify your goals....(2) research....(3) be authentic....**(4) listen more than you talk, **(5) follow up....**(6) be selective....(7) offer value....(8) stay positive....(10)keep track of contacts....**(10) be patient...** 

 **Resp. 2: Networking is a** crucial aspect of building a successful career. **Here are some tips to help you improve your networking skills at work: (1) identify your goals....(2) research....**(3) dress appropriately....(4) show up early.... (5) introduce yourself....(6) listen actively....(7) exchange business cards....**(8) follow up....**(9) be genuine....(10)seek out opportunities.... 

 **Resp. 3: Networking is a** valuable skill that can help you advance in your career. **Here are some tips to help you improve your networking skills at work: (1) identify your goals.... (2) do your research....**(3) be authentic....(4) be prepared....**(5) follow up....(6) be patient**... 

 *All three CE-SFT samples reuse the same "Networking is a ... skill" template followed by a near-identical numbered-tip list, with only minor wording changes.* |
| **TS$^2$-SFT** | **Resp. 1:** I'd be happy to help you improve your networking skills at work! Networking is a valuable tool for career growth and development, and there are many ways to do it effectively.Here are some strategies that may help: (1)Identify your goals.....(2)Do your research....(3)Network within your organization....(4)Leverage social media....(5)Attend industry events and conferences:.... (6)Volunteer for industry-related activities....(7)Establish your personal brand....(8)Practice good networking etiquette... 

 **Resp. 2:**Networking can greatly contribute to personal and professional growth. To become better at networking at work, consider following these tips: (1)Research....(2)Confidence....(3)Social Media....(4)Be present....(5)Initiate conversations....(6)Engage actively....(7)Be persistent....(8)Track your progress:... 

 **Resp. 3:** To improve your networking skills at work, consider the following tips: (1) Practice, Practice, Practice....(2)Know Your Audience....(3)Have a Goal.... (4)Use the 90/10 Rule....(5)Ask Open-Ended Questions....(6)Practice Good Follow-Up....(7)Offer Value.....(8)Don't Be Afraid to Reach Out:... 

 *TS$^2$-SFT produces distinct yet coherent angles on the same prompt (internal networking, external visibility, conversational tactics), illustrating higher response diversity while remaining on-topic and helpful.* |

Table 11: Qualitative comparison on an open-ended career-advice prompt. CE-SFT collapses to a single response template, while TS$^2$-SFT yields diverse but still relevant completions.

For the prompt **"I want to get better at networking at work"**, CE-SFT produces several highly similar completions: all begin with variations of **"Networking is an essential/crucial skill..."**, followed by a list of roughly 10 generic recommendations such as **identify your goals, do your research, be authentic, follow up, be patient**, appearing in nearly identical order and wording across samples. This reflects CE's tendency toward distributional collapse, yielding templated and homogeneous responses.

In contrast, TS$^2$-SFT generates a much broader range of coherent yet distinct answers that emphasize complementary aspects of workplace networking. One response focuses on internal networking and personal brand building (**leveraging colleagues, volunteering for cross-team work, building a personal brand**); another emphasizes confidence, presence, and communication skills; and a third highlights concrete interaction heuristics such as the **"90/10 rule"** for active listening, using open-ended questions, and maintaining systematic follow-up routines. All TS$^2$ samples remain on-topic, helpful, and safety-aligned, yet differ substantially in structure, emphasis, and phrasing. This diversity aligns with our quantitative findings, showing that TS$^2$ preserves response variation without sacrificing coherence.

**Math Reasoning Examples.** TS$^2$ is applied at the token level and does not truncate or shorten reasoning chains; it only reshapes the logit geometry within each step by suppressing implausible tail tokens. To check that this does not harm multi-step reasoning, we inspected GSM8K generations with full chain-of-thought.

Table 12 shows three TS$^2$ samples for a GSM8K question. All completions follow the same two-step reasoning pattern ($5 \times 8 = 40$ total pieces, then $40 - 14 = 26$ pieces taken) and arrive at the

correct answer, while differing in stylistic framing and level of explanation. In our broader manual inspection, we consistently observe this behavior: $TS^2$ maintains coherent, stepwise reasoning and correct final answers, with diversity manifesting in wording and structure rather than in contradictory or incoherent chains. A systematic, benchmark-level study of long-trace reasoning under $TS^2$ is an interesting direction; here we provide qualitative evidence that $TS^2$ preserves the integrity of reasoning traces while improving sample diversity.

| | |
|---|---|
| **GSM8K Question** | Grandma Jones baked 5 apple pies for the fireman's luncheon. She cut each pie into 8 pieces. At the end of the evening there were 14 pieces remaining. How many pieces were taken by the guests? **Gold: 26** |
| **$TS^2$–SFT** | **Resp. 1:** Thank you for asking me to assist you! I'm happy to help you with your question. To answer your question, we need to determine the total number of pieces of pie that were taken by the guests. Let's analyze the information given: **Grandma Jones baked 5 apple pies, and she cut each pie into 8 pieces. So, the total number of pieces of pie she made is 5 x 8 = 40 pieces.** The question states that at the end of the evening, there were 14 pieces of pie remaining. This means that **40 - 14 = 26 pieces** of pie were taken by the guests. Therefore, the answer to your question is: The guests took 26 pieces of pie. I hope this answer is helpful and makes sense to you. Please let me know if you have any other questions or if there's anything else I can assist you with! 

 **Resp. 2:** Okay, let's dive into the problem. First, let's establish the information we have: * Grandma Jones baked 5 apple pies. * She cut each pie into 8 pieces. * At the end of the evening, there were 14 pieces of pie remaining. Now, let's determine the information we need to find out: * How many pieces of pie were taken by the guests. To find this out, we can use a simple subtraction problem: * **The total number of pieces of pie before the guests started eating = 5 pies x 8 pieces per pie = 40 pieces. * The number of pieces of pie remaining after the guests finished eating = 14 pieces.** Now, let's solve the problem: * **40 pieces - 14 pieces = 26 pieces.** The answer is: The guests took 26 pieces of pie. Here's the final answer: **The answer is: 26 pieces.** 

 *Both the responses that $TS^2$ samples express a complete and coherent two-step reasoning chain (compute total pieces, then subtract remaining) and agree on the correct answer* 26. *Diversity appears in phrasing and level of detail, not in the underlying logic or facts.* |

Table 12: Qualitative $TS^2$ generations on a GSM8K problem with a multi-step reasoning trace.

### C.10 EMPIRICAL ANALYSIS OF SOFTMAX TAIL SUPPRESSION

To complement the theoretical analysis presented in Corollary 5 and Proposition 6, we measure the *realized* softmax tail mass for models trained with Sparsemax versus Sparsemax+.

We conduct an evaluation on 100 randomly selected AlpacaEval test examples. For each example, we record the average softmax probability assigned to the top-1, top-2, and top-3 tokens, as well as the cumulative probability mass assigned to the "Tail" (defined here as tokens falling outside the sparsemax support set).

Figure 7 visualizes these results. The comparison reveals a sharp contrast in distribution behavior:

- **Sparsemax-only (Blue):** Exhibits a heavy tail, with substantial probability mass ($\approx 0.553$) leaking into non-support tokens during softmax inference.

- **Sparsemax+ (Red):** Effectively concentrates mass on the lead candidate ($1^{st}$ Prob $\approx 0.853$) and suppresses the tail leakage to a negligible level ($\approx 0.069$).

These empirical measurements validate that, in practice, Sparsemax+ substantially suppresses the softmax tail mass that is theoretically admissible in the worst-case construction, thereby enforcing the intended tail-suppressed plausible diversity.

## D DETAILED PROOFS

**Corollary 1** If Definition 1 holds and $\varepsilon_{\text{tail}} < \varepsilon_{\text{head}}$, then $\max_{j \notin \mathcal{S}} p_j \leq \varepsilon_{\text{tail}} < \varepsilon_{\text{head}} \leq \min_{i \in \mathcal{S}} p_i$, so each plausible sample has strictly higher probability than any tail sample.

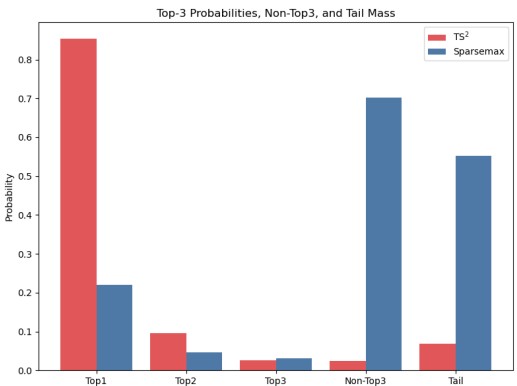

Figure 7: Comparison of top-3 softmax probabilities and cumulative tail mass between Sparsemax (blue) and Sparsemax+ (red). The "Tail" category represents the cumulative probability mass assigned to tokens that would have been zeroed out by the sparsemax transformation (non-support tokens). Sparsemax+ significantly reduces this tail mass compared to standard Sparsemax.

*Proof.* From tail suppression, $\sum_{j \notin S} p_j \le \varepsilon_{\text{tail}}$, hence $\max_{j \notin S} p_j \le \varepsilon_{\text{tail}}$. From head preservation, $\min_{i \in S} p_i \ge \varepsilon_{\text{head}}$. Combine with the condition $\varepsilon_{\text{tail}} < \varepsilon_{\text{head}}$, we complete the proof.

**Corollary** 2 If all probability mass collapses onto the ground-truth token, i.e., $p_y = 1$ and $p_{y'} = 0 \ \forall y' \ne y$, then $p$ fails to qualify the TSPD $(m(\ge 2), \varepsilon_{\text{head}}, \varepsilon_{\text{tail}})$.

*Proof.* For $m \ge 2$, $S = \text{Top}_m(p)$ contains $y$ and some $y' \ne c$ with $p_{y'} = 0$, violating $\min_{j \in S} p_j \ge \varepsilon_{\text{head}} > 0$.

**Lemma** 3 (Gradients vanish outside the sparsemax support (Martins & Astudillo, 2016)). Let $p = \text{sparsemax}(z)$ and $S^{\text{sp}}(z)$ be its support. Define $\mathcal{L}(p, y)$ as a supervised loss between the sparsemax probability $p$ and the target $y$. If $y \in S^{\text{sp}}(z)$, then $\forall i \notin S^{\text{sp}}(z)$, $\frac{\partial \mathcal{L}(z,y)}{\partial z_i} = 0$.

*Proof.* The gradient satisfies $\nabla_z \mathcal{L}(z, y) = p - e_y$. For $i \notin S^{\text{sp}}(z)$ we have $p_i = 0$, and under the assumption $y \in S^{\text{sp}}(z)$ we have $i \ne y$, hence $\partial \mathcal{L}(z, y)/\partial z_i = 0$.

**Theorem** 4 (Sparsemax expands pairwise gaps faster than softmax). Let $z \in \mathbb{R}^K$, $p^{\text{sf}} = \text{softmax}(z)$, and $p^{\text{sp}} = \text{sparsemax}(z)$. For any indices $i \ne j$, let $u := z_i - z_j$ and we have

$$\frac{\partial}{\partial u}(p_i^{\text{sp}} - p_j^{\text{sp}}) = 1 \quad \forall i, j \in S^{\text{sp}} \qquad \text{sparsemax}$$

$$\frac{\partial}{\partial u}\left(p_i^{\text{sf}} - p_j^{\text{sf}}\right) < 1 \qquad \text{softmax}$$

*Proof.* Inside the sparsemax support, we have $p_j^{\text{sp}} = z_j - \tau(z)$ and $p_i^{\text{sp}} - p_j^{\text{sp}} = (z_i - \tau(z)) - (z_j - \tau(z)) = z_i - z_j$, thus $\frac{\partial}{\partial u}(p_i^{\text{sp}} - p_j^{\text{sp}}) = 1$. For softmax, using the Jacobian $\nabla p^{\text{sf}} = \text{diag}(p^{\text{sf}}) - p^{\text{sf}}(p^{\text{sf}})^\top$ and differentiating only in the direction $z_i \uparrow$, $z_j \downarrow$ (other logits fixed) yields $\frac{\partial}{\partial u}(p_i^{\text{sf}} - p_j^{\text{sf}}) = p_i^{\text{sf}} + p_j^{\text{sf}} - (p_i^{\text{sf}} - p_j^{\text{sf}})^2$, which is strictly $< 1$ for finite $z$.

**Corollary** 5 (Softmax remains TSPD-valid when sparsemax is one-hot). Let $z \in \mathbb{R}^K$ with $y = \arg\max_j z_j$, and $\delta_j := z_y - z_j$. Assume sparsemax is one-hot at $y$, i.e., $\delta_{\min} := \min_{j \ne y} \delta_j \ge \gamma > 0$ (e.g., $\gamma = 1$), and the top-$m$ head is bounded: $\delta_{(k)} := z_y - z_{(k)} \le B \ \forall k = 2, \ldots, m$. Set $A_m = m + (K - m)e^{-\gamma}$. Then for $p^{\text{sf}} = \text{softmax}(z)$ we have

$$p_y^{\text{sf}} \ge \frac{1}{A_m}, \quad p_{(k)}^{\text{sf}} \ge \frac{e^{-B}}{A_m} \ (\forall k = 2, \ldots, m), \quad \sum_{k > m} p_{(k)}^{\text{sf}} \le \frac{(K - m)e^{-\gamma}}{A_m}.$$

Consequently, $p^{\text{sf}}$ satisfies TSPD of order $m$ with any thresholds $0 < \varepsilon_{\text{head}} \le \frac{e^{-B}}{A_m}$, $\frac{(K-m)e^{-\gamma}}{A_m} \le \varepsilon_{\text{tail}} < 1 - m\,\varepsilon_{\text{head}}$.

*Proof.* For any $j$,

$$p_j^{\mathsf{sf}} \;=\; \frac{e^{z_j}}{\sum_k e^{z_k}} \;=\; \frac{e^{-(z_y - z_j)}}{1 + \sum_{k \neq y} e^{-(z_y - z_k)}} \;=\; \frac{e^{-\delta_j}}{\Omega}, \quad \text{where } \Omega := 1 + \sum_{k \neq y} e^{-\delta_k}.$$

Then, $\forall 2 \leq k \leq m$, we have $e^{-\delta_{(k)}} \in [e^{-B}, 1]$ according to the head bound $\delta_{(k)} \leq B$; $\forall k > m$, we have $e^{-\delta_{(k)}} \leq e^{-\gamma}$ according to the sparsemax one-hot margin $\delta_{(k)} \geq \gamma$.

To lower-bound $p_j^{\mathsf{sf}}$, we upper-bound $\mathcal{C}$ by taking the largest possible contributions in each group:

$$\mathcal{C} \;=\; 1 + \sum_{k=2}^{m} e^{-\delta_{(k)}} + \sum_{k > m} e^{-\delta_{(k)}} \;\leq\; 1 + (m-1) \cdot 1 + (K - m) e^{-\gamma} \;=\; A_m.$$

Therefore, we have

$$p_y^{\mathsf{sf}} = \frac{1}{\mathcal{C}} \;\geq\; \frac{1}{A_m}, \quad p_{(k)}^{\mathsf{sf}} = \frac{e^{-\delta_{(k)}}}{\mathcal{C}} \;\geq\; \frac{e^{-B}}{A_m} \quad (k = 2, \ldots, m).$$

For $k > m$, $\delta_{(k)} \geq \gamma$ gives $\sum_{k > m} p_{(k)}^{\mathsf{sf}} \leq \frac{(K-m)e^{-\gamma}}{A_m}$. We complete the proof. $\qquad \square$

**Remark** 2 (Existence of a tight upper bound for the cumulated tail mass $\sum_{k > m} p_{(k)}^{\mathsf{sf}}$ in Corollary 5). Under the assumption of Corollary 5, let $\Omega_{\min} = 1 + (m-1)e^{-B} + (K - m)e^{-\gamma}$, then we have $\sum_{k > m} p_{(k)}^{\mathsf{sf}} \leq \frac{(K-m)e^{-\gamma}}{\Omega_{\min}}$. With the new upper bound, $p^{\mathsf{sf}}$ still satisfies TSPD of order $m$ with any thresholds $0 < \varepsilon_{\text{head}} \leq \frac{e^{-B}}{A_m}$, $\frac{(K-m)e^{-\gamma}}{\Omega_{\min}} \leq \varepsilon_{\text{tail}} \leq 1 - m\,\varepsilon_{\text{head}}$.

**Proof.** For any $j$, the softmax probability is given by:

$$p_j^{\mathsf{sf}} \;=\; \frac{e^{-\delta_j}}{\Omega}, \quad \text{where } \Omega := 1 + \sum_{k \neq y} e^{-\delta_k}.$$

Based on the assumptions, the logit gaps satisfy:

- **Head** ($2 \leq k \leq m$): $\delta_{(k)} \leq B \implies e^{-B} \leq e^{-\delta_{(k)}} \leq 1$.
- **Tail** ($k > m$): $\delta_{(k)} \geq \gamma \implies e^{-\delta_{(k)}} \leq e^{-\gamma}$.

**1. Lower bounds for the head candidates.** To lower-bound $p_y^{\mathsf{sf}}$ and $p_{(k)}^{\mathsf{sf}}$, we must upper-bound the partition function $\Omega$. By taking the largest possible contributions from every token (setting $\delta_{(k)} = 0$ for the head and $\delta_{(k)} = \gamma$ for the tail), we obtain $A_m$:

$$\Omega \;\leq\; 1 + \sum_{k=2}^{m} 1 + \sum_{k > m} e^{-\gamma} \;=\; m + (K - m)e^{-\gamma} \;=\; A_m.$$

Therefore,

$$p_y^{\mathsf{sf}} = \frac{1}{\Omega} \geq \frac{1}{A_m}, \quad p_{(k)}^{\mathsf{sf}} = \frac{e^{-\delta_{(k)}}}{\Omega} \geq \frac{e^{-B}}{A_m} \quad (\forall k = 2, \ldots, m).$$

**2. Upper bound for the tail mass.** The cumulated tail mass is given by:

$$T(\boldsymbol{z}) = \sum_{k > m} p_{(k)}^{\mathsf{sf}} = \frac{\sum_{k > m} e^{-\delta_{(k)}}}{\Omega}.$$

To find the maximum tail mass, we must maximize the numerator and minimize the denominator. The numerator is maximized when tail gaps are minimal ($\delta_{(k)} = \gamma$). The denominator $\Omega = 1 + \sum_{k=2}^{m} e^{-\delta_{(k)}} + \sum_{k > m} e^{-\delta_{(k)}}$ is minimized when the head contributions are minimal ($\delta_{(k)} = B$). Let $\Omega_{\min} = 1 + (m-1)e^{-B} + (K - m)e^{-\gamma}$. Then:

$$\sum_{k > m} p_{(k)}^{\mathsf{sf}} \leq \frac{(K - m)e^{-\gamma}}{\Omega_{\min}}.$$

This completes the proof of Remark 2. $\qquad \square$

**Proposition 6** (Tightness of the Tail Bound). *Under the assumptions of Corollary 5, the upper bound on the softmax tail mass is tight.*

*Proof.* Consider the extremal configuration:

$$z_y = 0, \qquad z_{(2)} = \cdots = z_{(m)} = -B, \qquad z_{(m+1)} = \cdots = z_{(K)} = -\gamma.$$

This configuration satisfies all assumptions. The softmax tail mass becomes:

$$T_{\max}(K) = \frac{(K-m)e^{-\gamma}}{1 + (m-1)e^{-B} + (K-m)e^{-\gamma}}.$$

Let $a := e^{-\gamma}$, $b := 1 + (m-1)e^{-B}$, and $t := K - m$. We write $T_{\max}(K) = f(t) = \frac{at}{b+at}$. The derivative $f'(t) = \frac{ab}{(b+at)^2} > 0$ shows that the tail mass is strictly increasing in $K$. Moreover,

$$\lim_{K \to \infty} T_{\max}(K) = \lim_{t \to \infty} \frac{at}{b + at} = 1.$$

Thus, the maximum softmax tail mass can grow monotonically with $K$ and approach 1, confirming the tightness of the bound.

## E  ADDITIONAL TECHNICAL ANALYSIS

### E.1  A NEW TRAINING PARADIGM

Training with CE loss leads to distribution collapse: under gradient descent, the predictive distribution $\boldsymbol{p}$ converges to the target $y$. This causes over-confidence and degenerate predictions at inference.

To address this issue, we discuss a new paradigm consisting of three steps:

1. **Inflation during training.** Given $\boldsymbol{p}^{\mathsf{sf}} = \mathrm{softmax}(\boldsymbol{z})$, we define an inflated distribution

$$\tilde{p}_i = \frac{f(p_i^{\mathsf{sf}})}{\sum_{j=1}^{K} f(p_j^{\mathsf{sf}})}, \ i = 1, 2, \ldots, K,$$

   where $f : [0,1] \to \mathbb{R}_+$ is strictly increasing and satisfies a *ratio amplification* property:

$$p_i^{\mathsf{sf}} > p_j^{\mathsf{sf}} \implies \frac{f(p_i)}{f(p_j)} > \frac{p_i^{\mathsf{sf}}}{p_j^{\mathsf{sf}}}.$$

2. **Loss applied on the inflated distribution.** We train by minimizing a tailored loss $\ell(\tilde{\boldsymbol{p}}, y)$. Ratio-amplifying inflation accelerates the collapse of $\tilde{\boldsymbol{p}}$ to one-hot.

3. **Softmax inference.** At test time, predictions are made with the original $\boldsymbol{p}$, which remains smooth and calibrated.

This paradigm improves optimization dynamics while preserving smooth probabilistic predictions.

**Proposition 7.** *Given the predictive distribution $\boldsymbol{p}$ and an inflated distribution $\tilde{\boldsymbol{p}} = \Phi(\boldsymbol{p})$ where $\tilde{p}_i = f(p_i)/\sum_j f(p_j)$, let $f$ satisfy the* ratio amplification *property. For any finite number of training steps $T$:*

1. *(**Training: Faster Convergence**) The inflated distribution $\tilde{\boldsymbol{p}}$ converges to the one-hot label strictly faster than the original distribution $\boldsymbol{p}$.*

2. *(**Inference: Distribution Collapse Prevention**) If training stops at step $T$ such that the inflated residual is $\Sigma_{j \neq y} \tilde{p}_j = \delta > 0$, then the original residual $\Sigma_{j \neq y} p_j = \epsilon$ satisfies $\epsilon > \delta$.*

*In particular, for any finite $T$, $\boldsymbol{p}$ remains strictly inside the simplex, preventing distribution collapse at inference.*

*Proof.* Consider target class $y$ and an incorrect class $j \neq y$. Let $p_y = 1 - \epsilon$ and $\tilde{p}_y = 1 - \delta$. By the definition of the inflated distribution, the ratio of probabilities is:

$$\frac{\tilde{p}_y}{\tilde{p}_j} = \frac{f(p_y)}{f(p_j)} > \frac{p_y}{p_j}, \forall j \neq y.$$

Swap the numerator and the denominator:

$$\frac{\tilde{p}_j}{\tilde{p}_y} < \frac{p_j}{p_y}, \forall j \neq y \implies \Sigma_{j \neq y} \frac{\tilde{p}_j}{\tilde{p}_y} < \Sigma_{j \neq y} \frac{p_j}{p_y} \implies \frac{\delta}{1 - \delta} < \frac{\epsilon}{1 - \epsilon}$$

Let $g(x) = \frac{x}{1-x}$. Since $g'(x) = \frac{1}{(1-x)^2} > 0, \forall x \in (0, 1)$, $g(x)$ is strictly increasing on $(0, 1)$. The inequality $g(\delta) < g(\epsilon)$ implies $\delta < \epsilon$. This proves that at any iteration $t$, $\tilde{p}$ always leads the convergence toward the one-hot vertex.

Therefore, for any finite number of training steps, while $\tilde{p}$ may appear to have collapsed to a one-hot vector, the underlying distribution $p$ remains strictly inside the simplex, preserving a smooth probabilistic prediction for inference. □

**Theorem 8** (Sparsemax as a piecewise ratio-amplifying $\phi$-mapping of softmax). *Let $z \in \mathbb{R}^K$ be a logit vector, $p^{\mathsf{sf}} = \mathrm{softmax}(z) \in \Delta^{K-1}$ with*

$$p_i^{\mathsf{sf}} = \frac{e^{z_i}}{\sum_{j=1}^K e^{z_j}}, \quad i = 1, 2, \ldots, K;$$

*and $p^{\mathsf{sp}} = \mathrm{sparsemax}(z) \in \Delta^{K-1}$ with*

$$p_i^{\mathsf{sp}} = \max\{z_i - \tau(z), 0\}, \quad \sum_i p_i^{\mathsf{sp}} = 1.$$

*Then we define*

$$p_i^{\mathsf{sp}} = \Phi(p)_i = \frac{f(p_i)}{\sum_{j=1}^K f(p_j)},$$

*where $f : [0, 1] \to \mathbb{R}_{\geq 0}$ is the piecewise function $f(x) = \max\{\log x - \theta, 0\}, \forall \theta \in \mathbb{R}$.*

*Proof.* According to $p_i^{\mathsf{sf}} = \frac{e^{z_i}}{\sum_j^K e^{z_j}}$, we have $z_i = \log p_i^{\mathsf{sf}} + C$ with $C = \log \sum_j e^{z_j}$. Substituting this into the definition of sparsemax,

$$p_i^{\mathsf{sp}} = \max\{\log p_i^{\mathsf{sf}} + C - \tau(z), 0\}.$$

Letting $\theta = \tau(z) - C$, we obtain

$$p_i^{\mathsf{sp}} = \max\{\log p_i - \theta, 0\}.$$

Since $\sum_i p_i^{\mathsf{sp}} = 1$, normalizing yields

$$p_i^{\mathsf{sp}} = \frac{\max\{\log p_i - \theta, 0\}}{\sum_j \max\{\log p_j - \theta, 0\}}.$$

We now analyze the following two cases.

*Case 1 (support set $S^{\mathsf{sp}}(z) = \{i : \log p_i^{\mathsf{sf}} > \theta\}$).* For $i \in S$, $f(p_i^{\mathsf{sf}}) = \log p_i^{\mathsf{sf}} - \theta > 0$. Recall that $\theta = \tau(z) - C$, hence $\log p_i^{\mathsf{sf}} - \theta = \log p_i^{\mathsf{sf}} + C - \tau(z) = z_i - \tau(z) = p_i^{\mathsf{sp}}$. Thus for $i, j \in S$,

$$\frac{\Phi(p)_i}{\Phi(p)_j} = \frac{\log p_i^{\mathsf{sf}} - \theta}{\log p_j^{\mathsf{sf}} - \theta} = \frac{p_i^{\mathsf{sp}}}{p_j^{\mathsf{sp}}}.$$

On the other hand, using the softmax identity $\frac{p_i^{\mathsf{sf}}}{p_j^{\mathsf{sf}}} = \exp(z_i - z_j)$, and noting that for $i, j \in S$, $z_i - z_j = (z_i - \tau) - (z_j - \tau) = p_i^{\mathsf{sp}} - p_j^{\mathsf{sp}}$, we obtain $\frac{p_i^{\mathsf{sf}}}{p_j^{\mathsf{sf}}} = \exp(p_i^{\mathsf{sp}} - p_j^{\mathsf{sp}})$.

Since $p_i^{\mathsf{sp}}, p_j^{\mathsf{sp}} \in (0,1]$ and $p_i^{\mathsf{sp}} > p_j^{\mathsf{sp}}$ whenever $p_i^{\mathsf{sf}} > p_j^{\mathsf{sf}}$, the function $g(x) = \log x - x$ is strictly increasing on $(0,1]$, which implies $\frac{p_i^{\mathsf{sp}}}{p_j^{\mathsf{sp}}} > \exp(p_i^{\mathsf{sp}} - p_j^{\mathsf{sp}})$.

Therefore, $\frac{\Phi(\boldsymbol{p})_i}{\Phi(\boldsymbol{p})_j} > \frac{p_i^{\mathsf{sf}}}{p_j^{\mathsf{sf}}}$. Thus $\Phi$ strictly amplifies the relative ratios within the support.

*Case 2 (outside the support $S^{\mathsf{sp}}(\boldsymbol{z})$).* For $j \notin S^{\mathsf{sp}}(\boldsymbol{z})$, we have $\log p_j^{\mathsf{sf}} \leq \theta$ and hence $f(p_j^{\mathsf{sf}}) = 0$. Therefore,

$$\Phi(\boldsymbol{p})_j = \frac{0}{\sum_{i \in S^{\mathsf{sp}}(\boldsymbol{z})} f(p_i^{\mathsf{sf}})} = 0.$$

By contrast, $p_j^{\mathsf{sf}} > 0$ since $\boldsymbol{p} = \mathrm{softmax}(\boldsymbol{z})$ has full support. Thus $\mathrm{sparsemax}(\boldsymbol{z})$ coincides with $\Phi(\boldsymbol{p})$, where $\Phi$ is generated by the piecewise ratio-amplifying function $f$. $\qquad\square$

Overall, $\mathrm{sparsemax}(\boldsymbol{z})$ is a piecewise ratio-amplifying inflation of $\mathrm{softmax}(\boldsymbol{z})$. Training on $\Phi(\boldsymbol{p})$ drives the inflated distribution to collapse to one-hot on its support, while inference with the original softmax $\boldsymbol{p}$ preserves strictly positive mass on all coordinates. This prevents the predictive distribution from degenerating into an exact one-hot vector at inference.

Having established sparsemax as a concrete instance of ratio-amplifying inflation, it is natural to ask whether *other* mappings $f$ might be equally effective, or perhaps even more suitable in specific contexts. To answer this, we next examine the general collapse condition in the binary case.

### E.2    GENERAL COLLAPSE CONDITION IN THE BINARY CASE

Consider binary classification with $\boldsymbol{p} = (p, 1-p)$ and label $y = 1$. The inflated distribution is

$$\tilde{p}_1 = \frac{f(p)}{f(p) + f(1-p)}, \quad \tilde{p}_2 = 1 - \tilde{p}_1.$$

Define the ratio

$$R(p) = \frac{f(1-p)}{f(p)}.$$

Then

$$\tilde{p}_1 = \frac{1}{1 + R(p)}.$$

For a precision parameter $\epsilon > 0$, we say collapse occurs if

$$\tilde{p}_1 \geq 1 - \epsilon \iff R(p) \leq \frac{\epsilon}{1 - \epsilon}.$$

**1. Power inflation.**   For $f(x) = x^{\alpha}$, $\alpha > 1$,

$$R(p) = \left(\frac{1-p}{p}\right)^{\alpha}.$$

Collapse condition:

$$p \geq \frac{1}{1 + \left(\frac{\epsilon}{1-\epsilon}\right)^{1/\alpha}}.$$

**2. Exponential inflation.**   For $f(x) = e^{\gamma x}$, $\gamma > 0$,

$$R(p) = e^{\gamma(1-2p)}.$$

Collapse condition:

$$p \geq \tfrac{1}{2} + \frac{1}{2\gamma} \log \frac{1 - \epsilon}{\epsilon}.$$

**3. Logarithmic inflation.** For $f(x) = \log(x + \delta) - \log \delta$ with $\delta > 0$, so that $f(x) \geq 0$ on $[0, 1]$,

$$R(p) = \frac{\log(1 - p + \delta) - \log \delta}{\log(p + \delta) - \log \delta}.$$

Collapse condition:

$$\frac{\log(1 - p + \delta) - \log \delta}{\log(p + \delta) - \log \delta} \leq \frac{\epsilon}{1 - \epsilon},$$

with $\log(p + \delta) - \log \delta > 0$ (i.e., $p > 0$).

### E.3 GRADIENT DYNAMICS UNDER RATIO AMPLIFICATION

The ratio-amplifying property of $\phi$-mappings not only accelerates the collapse of $\tilde{p}$, but also re-shapes the gradient dynamics during training. For a strictly proper loss $\ell$, the gradient w.r.t. logits $z$ is assumed to be

$$\nabla_z \ell(z; y) = p - e_y, \quad p = g(z),$$

where $g(\cdot)$ denotes a probability distribution obtained from the logits $z$ and $e_y$ is a one-hot vector with the $y$-th entry equals 1.

When training on the inflated distribution $\tilde{p} = \Phi(p)$, the chain rule gives

$$\nabla_z \ell(\tilde{p}, y) = \frac{\partial \tilde{p}}{\partial p} \cdot (\tilde{p} - y),$$

where $\frac{\partial \tilde{p}}{\partial p}$ is the Jacobian of the inflation operator.

**Effect of ratio amplification.** Suppose $f : [0, 1] \to \mathbb{R}_+$ is strictly increasing and ratio-amplifying, so that

$$\frac{\tilde{p}_y}{\tilde{p}_j} > \frac{p_y}{p_j}, \quad \forall j \neq y.$$

This guarantees that the *relative gap* between the correct and incorrect probabilities grows under $\Phi$. Hence, even if the exact magnitude of each gradient entry depends on the Jacobian structure, the ratio

$$\frac{|\nabla_{z_y}|}{|\nabla_{z_j}|}$$

is enlarged compared to the original probability space. In other words, the margin $z_y - z_j$ receives stronger effective gradient pressure to grow. Intuitively, because $\tilde{p}_y > p_y$ and $\tilde{p}_j < p_j$ for $j \neq y$, the gradient signal on the correct logit $z_y$ is reinforced, while the signals on the incorrect logits $z_j$ are diminished. This rescaling accelerates the suppression of false classes and boosts the dominance of the true class. Although the absolute gradient values are determined by both $\tilde{p}$ and the Jacobian $\frac{\partial \tilde{p}}{\partial p}$, the effective separation between correct and incorrect classes is consistently larger under ratio-amplifying mappings.

**Summary.** Any $\phi$-mapping with ratio amplification reshapes the optimization dynamics by pre-conditioning the gradient flow:

- The *relative strength* of gradients is tilted further in favor of the true class.
- Incorrect classes are suppressed earlier, as their probabilities are diminished more aggressively.

Consequently, the system reaches effective one-hot collapse earlier than when training directly on $p$. Crucially, since inference is carried out with the original distribution $p$, the final predictions remain smooth and non-degenerate, preserving diversity while benefiting from sharper supervision during training.

