# OpenReview forum: "TS$^2$: Training with Sparsemax+, Testing with Softmax for Accurate and Diverse LLM Fine-Tuning"
_ICLR.cc/2026/Conference — ICLR 2026 Poster_

### Official Review · Reviewer_jH36 · 2025-10-29

**Soundness:** 4
**Presentation:** 4
**Contribution:** 3
**Rating:** 6
**Confidence:** 3

**Summary:**

The paper proposes an alternative to the cross-entropy loss for language model SFT. The main motivation for this is that CE loss pushes all the probability mass under the golden token, killing the diversity in the pre-trained model. The proposed method uses sparse activation (sparsemax+) to train the model on only plausible tokens. This makes the model output diverse when used with softmax while keeping it accurate. The paper shows better diversity and accuracy compared to existing methods like GEM, CE+weighting (CE regularization), and CE.

**Strengths:**

- The results show strong performance and diversity on a variety of benchmarks and are measured on a variety of metrics for diversity.
- Results show that better diversity also leads to sample efficiency when doing BON
- Ablation studies show holding out one of the component leads to drop in performance, indicating the need for all components (sparsemax+, softmax decoding) for the best performance
- The method is compared against existing SOTA (GEM)
- The motivation is clearly laid out, and the research questions are made clear

**Weaknesses:**

- The paper discusses the overconfidence due to the SFT issue, but doesn't discuss existing work on it
- Other relevant methods to prevent diversity collapse are mentioned, but don't have a comparison against (Label smoothing, unlikelihood)

**Questions:**

- For baselines, did you consider label smoothing and α-Entmax since they directly address the overconfidence problem?
- I am curious to know the effects of the TS2 loss on factual accuracy when the diversity is high? As in, when you sample multiple times, do the facts stay consistent?
- Can authors comment on the effectiveness of this method when training on datasets with long reasoning traces? Does the model still give coherent reasoning steps when diversity is maintained?
- Similar to above, how does it perform on a multi-turn dataset (for example, on tau-bench)?
- Could you share the decoding hyperparameters? Since this could influence the measured diversity

---

> ### Author Response · Authors · 2025-11-22
> **Response to Reviewer jH36**
>
> We thank the reviewer for valuable comments. We address the concerns in detail below.
>
> > **W1:** The paper discusses the overconfidence due to the SFT issue, but doesn't discuss existing work on it.
>
> **Response:** Thank you for pointing this out. Our current draft primarily frames overconfidence through the lens of diversity loss and cites decoding-time heuristics and training-time regularizers such as label smoothing and unlikelihood training, as well as GEM. We agree that it would be clearer to explicitly connect this discussion to the calibration / overconfidence literature on (pre-trained) language models.
>
> In the revision, we have add a dedicated paragraph in the **Related Work** section discussingelated-work section summarizing prior work that studies miscalibration and overconfidence in neural LMs and LLMs, including classical deep network calibration and recent work on language model calibration and confidence elicitation.
>
> ---
>
> > **W2&Q1:** For baselines, did you consider label smoothing and $\alpha$-Entmax since they directly address the overconfidence problem?
>
> **Response:** Thanks for pointing this out. We added the comparison with label smoothing and $\alpha$-Entmax as below (also included as **Appendix C.2 Table 4** in the manuscript).
>
> | Metric                | TS$^2$ (ours) | GEM   | CE    | CE + label smoothing | sparsemax | 1.5-Entmax |
> | :-------------------- | :-------------- | :---- | :---- | :------------------- | :-------- | :--------- |
> | **Winrate (%)** | **33.12** | 31.53 | 29.77 | 28.25                | 12.87     | 13.51      |
>
> *Table 1: Comparison of TS$^2$, GEM, CE, sparsemax, Entmax, and CE with label smoothing on AlpacaEval (Chat).*
>
> We observe that while CE + label smoothing and 1.5-Entmax increase diversity, they do so at a substantial cost in win-rate and remain clearly below both GEM and TS$^2$. Sparsemax and Entmax also lack an explicit mechanism to control the cumulative tail mass: they either over-flatten the distribution (label smoothing) or produce sparse supports without directly suppressing the softmax tail at inference, which hurts helpfulness and alignment.
>
> In contrast, TS$^2$ combines a sparse training mapping with an explicit tail-suppression term and softmax decoding. This preserves or improves win-rate (highest among all baselines) while still delivering strong diversity, indicating a more favorable trade-off between confidence, diversity, and task performance than these overconfidence-oriented baselines.
>
> ---
>
> > **Q2:** I am curious to know the effects of the TS$^2$ loss on factual accuracy when the diversity is high? As in, when you sample multiple times, do the facts stay consistent?
>
> **Response:** TS$^2$ increases diversity by reshaping the logit geometry, not by injecting randomness into low–probability tails. Sparsemax+ sharpens the margin between plausible and implausible tokens and *suppresses* out-of-support mass; at inference, we still decode with softmax over the same logits. Thus, the high–probability, fact-bearing continuations of the base model remain dominant, while diversity comes from exploring multiple plausible modes rather than from sampling unstable tail tokens.
>
> Empirically, in our multi-sample inspection, we do not observe TS$^2$ introducing contradictory factual statements. The qualitative example can be seen in **Table 2 of the response to reviewer `uXcp`** (better view in **Appendix C.9** **Table 11**). It shows different responses emphasizes a different aspect of workplace networking (internal relationships and events, external visibility and social media, conversational tactics), yet all maintain consistent, non-conflicting advice. Across other prompts, we find that TS$^2$ mainly changes *framing and focus* while preserving the underlying factual content of the base model.
>
> A more systematic, benchmark-style study of factual consistency under multi-sampling is an interesting direction; we will clarify in the revision that our current focus is on diversity and calibration, and leave a dedicated factuality analysis to future work.

---

> > ### Author Response · Authors · 2025-11-22
> > **Response to Reviewer jH36 (continued)**
> >
> > > **Q3:** Can authors comment on the effectiveness of this method when training on datasets with long reasoning traces? Does the model still give coherent reasoning steps when diversity is maintained?
> >
> > **Response:** TS$^2$ is applied at the token level and does not truncate or shorten reasoning chains; it only reshapes the logit geometry within each step by suppressing implausible tail tokens. To check that this does not harm multi-step reasoning, we inspected GSM8K generations with full chain-of-thought.
> >
> > The table below (detailed in **Appendix C.9** **Table 12**) shows TS$^2$ samples for a GSM8K question. The responses follow the same two-step reasoning pattern ($5\times 8 = 40$ total pieces, then $40-14 = 26$ pieces taken) and arrive at the correct answer, while differing in stylistic framing and level of explanation. In our broader manual inspection, we consistently observe this behavior: TS2 maintains coherent, stepwise reasoning and correct final answers, with diversity manifesting in wording and structure rather than in contradictory or incoherent chains.
> >
> > | GSM8K Question | Grandma Jones baked 5 apple pies for the fireman's luncheon. She cut each pie into 8 pieces. At the end of the evening there were 14 pieces remaining. How many pieces were taken by the guests? `Gold : 26` |
> > | :-| :- |
> > | TS$^2$–SFT | **Resp. 1:** Thank you for asking me to assist you! I'm happy to help you with your question. To answer your question, we need to determine the total number of pieces of pie that were taken by the guests. Let's analyze the information given: `Grandma Jones baked 5 apple pies, and she cut each pie into 8 pieces. So, the total number of pieces of pie she made is 5 x 8 = 40 pieces.` The question states that at the end of the evening, there were 14 pieces of pie remaining. This means that `40 - 14 = 26 pieces` of pie were taken by the guests. Therefore, the answer to your question is: The guests took 26 pieces of pie. I hope this answer is helpful and makes sense to you. Please let me know if you have any other questions or if there's anything else I can assist you with! **Resp. 2:** ... |
> >
> >
> > *Table 1: Qualitative TS$^2$ generations on a GSM8K problem with a multi-step reasoning trace. See Appendix Table 12 for more responses.*
> >
> > ---
> >
> > > **Q4:** Similar to above, how does it perform on a multi-turn dataset (for example, on tau-bench)?
> >
> > **Response:** Thanks for this question. We evaluated our method on **MT-Bench-101** (scored on 0~10), comparing CE, GEM and TS$^2$. This benchmark contains fine-grained metrics to evaluate the performance of models specifically in multi-turn dialogue settings. This is present as **Table 5** in the **Appendix C.3** section of the manuscript.
> >
> > Our method shows consistent improvements across all multi-turn capability groups (e.g., +0.95 Context Memory, +1.34 Content Confusion). These results indicate that our method performs robustly in multi-turn dialogue settings.
> >
> > | Model                    | `Avg.`       | Memory (CM)    | Understanding (SI / AR) | Interference (TS / CC) | Rephrasing (CR / FR)  | Reflection (SC / SA)           | Reasoning (MR / GR)  | Questioning (IC / PI) |
> > | :----------------------- | :------------- | :------------- | :---------------------- | :--------------------- | :-------------------- | :----------------------------- | :------------------- | :-------------------- |
> > | **CE**             | 5.99           | 5.01           | 4.59 / 6.03             | 5.10 / 5.03            | 7.33 / 7.02           | 6.34 / 7.38                    | 6.37 / 4.67          | 7.49 / 5.46           |
> > | **GEM**            | 6.24           | 4.63           | 4.43 / 6.88             | 5.95 / 5.36            | 7.19 / 7.76           | 7.20 / 8.05                    | 6.09 / 5.05          | 6.54 / 5.98           |
> > | **TS$^2$(Ours)** | **6.65** | **5.96** | **4.76** / 6.58   | 5.94 /**6.70**   | **8.27** / 6.60 | **7.42** /**8.53** | 6.03 /**6.14** | 6.80 /**6.76**  |
> >
> > *Table 2: Comparison across MT-Bench-101 capability dimensions. See **Appendix Table 5** for the detailed description for the evaluation metrics.*
> >
> > ---
> >
> > > **Q5:** Could you share the decoding hyperparameters? Since this could influence the measured diversity.
> >
> > **Response:** We provide the details here. We utilized **vLLM** as our inference backend for all experiments, with the random seed fixed at 42. Regarding the sampling strategy, we set the temperature to 0.6, combined with **top-$p$** (nucleus) sampling at 0.9 and **top-$k$** filtering at 50. We generated **$N=32$** independent samples for each input prompt to estimate the diversity metrics, with a maximum generation length restricted to 2048 tokens.

---

### Official Review · Reviewer_uXcp · 2025-11-01

**Soundness:** 3
**Presentation:** 3
**Contribution:** 3
**Rating:** 6
**Confidence:** 4

**Summary:**

This paper addresses the issue of output diversity loss in supervised fine-tuning (SFT) of large language models. Standard SFT with cross-entropy (CE) tends to over-fit the model's output distribution to one-hot targets, suppressing alternative correct continuations and thus reducing generative diversity. The authors propose TS$^2$ (Training with Sparsemax+, Testing with Softmax) as a drop-in replacement for the usual fine-tuning procedure. During training, they replace the softmax+CE loss with a modified Sparsemax+ loss, which encourages a sparse probability distribution over next-token outputs while adding a suppression term to push down probabilities of tokens outside the support of the target. At inference, the model still uses a normal softmax for decoding. This strategy preserves plausible near-tie candidates in the output distribution (instead of forcing a single high-probability token as CE does) while still penalizing irrelevant tokens. The paper's contributions include: (1) a new Sparsemax+ transformation and loss function that fixes a known issue of vanilla sparsemax (which left too much residual probability on tail classes), and (2) demonstrating that TS$^2$ yields consistent improvements in both accuracy and output diversity over the baseline CE fine-tuning. The results suggest that TS$^2$ is an effective, easy-to-implement approach to obtain LLMs that are more accurate while also more creative (producing varied outputs).

**Strengths:**

* **Novelty:** The paper tackles the problem of maintaining output diversity in supervised fine-tuning. It introduces a creative solution by leveraging sparsemax (a non-standard probability transform) in a new way. The proposed Sparsemax+ improves upon sparsemax by addressing its known gradient masking issue for out of support set classes. This combination (train with sparsemax+, test with softmax) is a novel idea that, to my knowledge, has not been explored in prior LLM fine-tuning literature.

* **Quality:** The authors fine-tune two open models (Llama 3.1 8B and Qwen 2.5 7B) on multiple benchmarks covering different domains (chat/dialog, code generation, and open-domain tasks). The results consistently show improvements over the baseline cross-entropy fine-tuning. Importantly, both accuracy (task performance) and diversity metrics improve, which demonstrates the method's effectiveness without trade-offs. The paper also provides intuition (and some theoretical discussion) for why TS$^2$ works, grounding the empirical findings.

* **Clarity:** The paper is clearly written with a logical flow. The authors motivate the problem (lack of alternative continuations under CE) convincingly, and then explain their solution step-by-step. Key terms like sparsemax and the new loss are defined, and the intuition is easy to grasp.

* **Significance:** The findings have practical significance for the field. Many applications of LLMs (creative writing, coding assistant, dialogue systems) benefit from diverse outputs, and TS$^2$ provides a simple drop-in technique to achieve that without sacrificing accuracy. This is significant because typically one might expect a trade-off between diversity and correctness. The method is also easy to implement for practitioners (just changing the loss function during training), which means it could be adopted widely if these results hold.

**Weaknesses:**

* **Missing Baseline Comparisons:** The experimental section might be missing some baselines for a fair assessment. In particular, it's important to know how TS$^2$ compares against simpler fixes to the one-hot problem, such as label smoothing during CE training or using entmax loss [1] (which is a family that includes sparsemax and softmax). If those were not evaluated, the authors should clarify why. Including such baselines would strengthen the paper by demonstrating that Sparsemax+ indeed outperforms these alternatives in balancing accuracy and diversity. Without them, it's harder to quantify the gain from the new method versus existing tricks.

* **Impact on Downstream Use Cases:** The paper shows benchmark improvements, but have the authors examined qualitative examples or specific scenarios illustrating the benefit? For instance in a chat setting, does TS$^2$ produce more varied and interesting responses to open-ended questions (compared to CE which might produce more templated answers)? A few concrete examples in the rebuttal version would be great to demonstrate the practical difference in model behavior.

* **Hyperparameter Sensitivity:** How sensitive is the Sparsemax+ training to its hyperparameter - the strength of the out-of-support suppression term $\alpha$? The paper mentions that $\alpha$ is empirically determined for each model architecture, but additional analysis on its effect (example varying $\alpha$ to show robustness or convergence behavior) would help readers better understand how critical this choice is and how generalizable the method is.

[1] "Sparse Sequence-to-Sequence Models", Ben Peters and Vlad Niculae and André F. T. Martins, https://arxiv.org/abs/1905.05702.

**Questions:**

The questions are the same as those listed in the Weaknesses section.

---

> ### Author Response · Authors · 2025-11-22
> **Response to Reviewer uXcp**
>
> We thank the reviewer for valuable comments. We address the concerns in detail below.
>
> > **W1:** Missing Baseline Comparisons.
>
> **Response:** Thank you for your constructive comment. We have now evaluated both label smoothing and entmax loss baselines.
>
> For **label smoothing**, the mechanism reallocates part of the target probability mass uniformly across all non-target tokens. Although the probability of alternative tokens increases, it indiscriminatingly boosts the probability of nonsensical or semantically irrelevant tokens alongside plausible ones. This indiscriminate diversity proved detrimental to generation quality. Additionally, given the large vocabulary size (e.g., Llama-3.1-8b has ~120k+ tokens), the smoothing mass is diluted so thinly that the practical lift in plausible diversity is negligible.
>
> As for **$\alpha$-entmax loss**, we tested it (specifically 1.5-entmax) and observed issues similar to standard sparsemax loss: it struggles to penalize the tail sufficiently, often maintaining too much probability mass on irrelevant tokens, which degrades performance during softmax inference.
>
> We have trained and tested the exact model used in the paper with label smoothing and entmax loss. The results are shown in the table below (present as **Table 4** in **Appendix C.2**). TS$^2$ significantly outperforms both baselines.
>
> | Metric            | TS$^2$(Ours)  | GEM   | CE    | CE+Label Smoothing | Sparsemax | 1.5-Entmax |
> | :---------------- | :-------------- | :---- | :---- | :----------------- | :-------- | :--------- |
> | **Winrate** | **33.12** | 31.53 | 29.77 | 28.25              | 12.87     | 13.51      |
>
> *Table 1: Comparison of TS$^2$, GEM, CE, sparsemax, entmax, and CE with label smoothing on AlpacaEval.**
>
>
> > **W2:** Impact on Downstream Use Cases.
>
> **Response:** We have added a qualitative comparison in **Appendix C.9** (**Table 11**).
>
> For the prompt *"I want to get better at networking at work"*, CE-SFT produces three highly similar completions: all begin with *"Networking is an essential/crucial skill..."*, then list ~10 generic tips (*identify your goals, do your research, be authentic, follow up, be patient*), in almost identical order and wording.
>
> By contrast, TS$^2$-SFT generates diverse yet coherent answers that emphasize complementary aspects of the same task: one focuses on internal networking and personal brand (leveraging colleagues, volunteering, and establishing your personal brand), another stresses confidence and communication skills, and a third highlights concrete interaction heuristics (*"90/10 rule"* for listening, open-ended questions, systematic follow-up). All remain on-topic, helpful, and safety-aligned, but differ in structure, emphasis, and phrasing.
> | Prompt | I want to get better at networking at work. |
> | :----- | :------------------------------------------- |
> | **CE-SFT**  | **Response 1:** `Networking is an` essential skill for personal and professional growth. `Here are some tips to help you improve your networking skills at work: (1) identify your goals....(2) research....(3) be authentic....` (4) listen more than you talk, `(5) follow up....` (6) be selective.... (7) offer value.... (8) stay positive.... (10) keep track of contacts.... `(10) be patient....` **Response 2:** `Networking is a` crucial aspect of building a successful career. `Here are some tips to help you improve your networking skills at work: (1) identify your goals....(2) research....`(3) dress appropriately.... (4) show up early.... (5) introduce yourself.... (6) listen actively.... (7) exchange business cards....* `(8) follow up....` (9) be genuine.... (10) seek out opportunities.... **Response 3:** `Networking is a` valuable skill that can help you advance in your career. .... |
> | **TS2-SFT** | **Response 1:** I'd be happy to help you improve your networking skills at work! Networking is a valuable tool for career growth and development, and there are many ways to do it effectively. Here are some strategies that may help: (1) Identify your goals..... (2) Do your research.... (3) Network within your organization.... (4) Leverage social media.... (5) Attend industry events and conferences:.... (6) Volunteer for industry-related activities.... (7) Establish your personal brand.... (8) Practice good networking etiquette... **Response 2:** Networking can greatly contribute to personal and professional growth. To become better at networking at work, consider following these tips: (1) Research.... (2) Confidence.... (3) Social Media.... (4) Be present.... (5) Initiate conversations.... (6) Engage actively.... (7) Be persistent.... (8) Track your progress:... **Response 3:** To improve your networking skills at work, ...
>
> *Table 2: Qualitative comparison on an open-ended career-advice prompt. Better view in Appendix Table 11 for more details.*

---

> > ### Author Response · Authors · 2025-11-22
> > **Response to Reviewer uXcp (continued)**
> >
> > > **W3:** Hyperparameter Sensitivity.
> >
> > **Response:** We conducted a targeted ablation on the Sparsemax+ hyperparameter $\alpha$ for both LLaMA-3.1-8B and Qwen2-7B.
> >
> > For LLaMA-3.1-8B on AlpacaEval (see table below, also added as **Table 3** in **Appendix C.1**), we vary $\alpha \in \{0.25, 0.50, 0.75, 1.0, 5.0\}$ and observe:
> >
> > * **Smooth behavior for moderate $\alpha$:** In the range $\alpha \in [0.25, 1.0]$, win rate and diversity metrics (n-gram, self-BLEU, Sent-BERT) change smoothly rather than abruptly. The best trade-off is achieved at $\alpha = 0.25$: 33.12% win rate with strong diversity.
> > * **Over-suppression for very large $\alpha$:** When $\alpha$ is increased to 5.0, the tail suppression term dominates, leading to substantially lower win rate (25.34%) and reduced diversity, confirming the intuition that excessively penalizing out-of-support mass hurts performance.
> >
> > | Setting         | Winrate (%)     | n-gram          | 100-self-BLEU   | Sent-BERT       |
> > | :-------------- | :-------------- | :-------------- | :-------------- | :-------------- |
> > | $\alpha$=0.25 | **33.12** | **23.78** | **53.87** | **12.80** |
> > | $\alpha$=0.50 | 31.44           | 20.54           | 51.34           | 11.61           |
> > | $\alpha$=0.75 | 30.84           | 19.07           | 49.29           | 10.96           |
> > | $\alpha$=1.00 | 30.39           | 18.13           | 48.05           | 10.30           |
> > | $\alpha$=5.00 | 25.34           | 13.44           | 41.42           | 8.12            |
> >
> > *Table 3: Effect of $\alpha$ on Chat (AlpacaEval).*
> >
> > Similarly for the Qwen model, the optimal value was found to be 0.5. We also note that training is stable across all tested settings; varying $\alpha$ mainly trades off win rate versus diversity rather than causing convergence issues. In practice, we find that choosing $\alpha$ from a moderate range such as $[0.25, 0.75]$ transfers well between models with similar logit scales, suggesting that Sparsemax+ is not overly sensitive to this hyperparameter.

---

### Official Review · Reviewer_2nRT · 2025-11-01

**Soundness:** 2
**Presentation:** 2
**Contribution:** 2
**Rating:** 4
**Confidence:** 4

**Summary:**

This paper proposes TS2 (Train with Sparsemax+, Test with Softmax), a novel fine-tuning framework for large language models that aims to mitigate distributional collapse and preserve useful diversity. The key idea is to decouple the probability mappings used in training and inference. During training, the model uses Sparsemax+, an enhanced version of sparsemax that includes a tail-suppressing penalty to eliminate residual probability mass on implausible tokens, thereby enforcing “Tail-Suppressed Plausible Diversity (TSPD).” At inference, the model reverts to the softmax mapping to restore smooth, calibrated probabilities, allowing exploration among plausible alternatives. The authors provide theoretical analysis on gradient masking and sparsity behavior, proving that sparsemax expands pairwise logit gaps faster than softmax and yields a more decisive support set. Empirically, TS2 is benchmarked against CE, NEFT, and GEM on both chat and code-generation tasks, showing consistent improvements in win rate, pass@k, and diversity metrics. Ablation studies further confirm that the combination of sparsemax training, softmax inference, and suppression penalty is crucial to the method’s success.

**Strengths:**

Empirically, the results are comprehensive and convincing, spanning multiple domains (chat, code, creative writing, and reasoning benchmarks) and multiple base models (Llama-3.1-8B, Qwen-2-7B).

The method outperforms strong baselines such as GEM not only in accuracy but also in diversity metrics, demonstrating improvement both in quality and variety. The clarity of presentation and reproducibility documentation is also commendable.

**Weaknesses:**

1. The main limitation is the upper bound of the cumulated tail mass of softmax outside the top-$m$,
$$
\sum_{k > m} p^{\mathrm{sf}}_{(k)} \le \frac{(K - m)e^{-\gamma}}{A_m}.
$$
It is said that this upper bound is strictly increasing in $K$ (for fixed $m, \gamma$) and approaches $1$ as $K \to \infty$. Finally, the authors state that ''with large vocabularies, the admissible tail under softmax at inference becomes non-negligible.''

However, the statement is not support by the theorem. The upper bounds goes to 1 does not tell anything about the probability. To support the claim, the authors should prove that the upper bound is tight, or provide a lower bound on the probability.

2. The proposed loss is simple, compressing the tailed probability to 1 single probability. To support that the proposed one is better, the authors should provide a result like Corollary 5 with a tighter bound (if possible). Otherwise, it is hard to tell whether the proposed loss resolved the raised theoretical issue.

Writing and presentation issues:

1. The definition of sparsemax (lines 154–157) appears to be ill-defined. Specifically, the threshold function is defined in terms of the support set, yet the support set itself is defined based on the threshold function, resulting in an apparent circular definition. I suspect that sparsemax has a formal, non-circular definition in its original paper, and that the circular definition presents only two derived properties rather than the definition itself.

2. Lemma 3, Lsp(z,y) is not defined before.Leaving readers to guess whether it is refered to “we train with sparsemax and optimize a modified Fenchel–Young loss tailored to this mapping” mentioned in the introduction.

**Questions:**

see weakness

---

> ### Author Response · Authors · 2025-11-22
> **Response to Reviewer 2nRT**
>
> We thank the reviewer for valuable comments. We address your comments in detail below.
>
> > **W1:** The main limitation is the upper bound of the cumulated tail mass of softmax outside the top-$m$, $\sum_{k>m} p^{\mathrm{sf}}_{(k)} \le\frac{(K-m)e^{-\gamma}}{A_m}$. ... The upper bounds going to 1 does not tell anything about the probability. To support the claim, the authors should prove that the upper bound is tight, or provide a lower bound on the probability.
>
> **Response:** Thank you for the insightful comment. We agree that the fact that an *upper* bound tends to 1 as $K \to \infty$ does not, by itself, imply that the realized softmax tail mass is large in practice. Our statement that “the admissible tail under softmax becomes non-negligible for large vocabularies” is intended in a *worst-case* sense: it characterizes how large the tail *can* be under the assumptions of Corollary 5, not how large it typically is.
>
> * **Existence of a tight upper bound for the cumulated tail mass $\sum_{k>m} p_{(k)}^{\mathrm{sf}}$ in Corollary 5.**
>   Let $\Omega_{\min} = 1 + (m-1)e^{-B} + (K-m)e^{-\gamma}$, then we have $\sum_{k>m} p_{(k)}^{\mathrm{sf}} \le\frac{(K-m)e^{-\gamma}}{\Omega_{\min}}$. With the new upper bound, $p^{\mathrm{sf}}$ still satisfies TSPD of order $m$ with any thresholds $0<\varepsilon_{\mathrm{head}} \le  \frac{e^{-B}}{A_m}$, $\frac{(K-m)e^{-\gamma}}{\Omega_{\min}}\le\varepsilon_{\mathrm{tail}}\le1-m\varepsilon_{\mathrm{head}}$ (see Appendix D for detailed derivation).
>   Furthermore, the upper bound $\Omega_{\min}$  on the cumulated tail mass $\sum_{k>m} p_{(k)}^{\mathrm{sf}}$ is strictly increasing in $K$ and approaches $1$ as $K \to\infty$. Importantly, this bound is tight. See Proposition 6 in Appendix D: there exists a worst-case configuration $z_y = 0, z_{(2)} = \cdots = z_{(m)} = -B, z_{(m+1)} = \cdots = z_{(K)} = -\gamma$, where the tail mass grows monotonically with $K$. Thus, for large vocabularies, the admissible tail under softmax at inference becomes nearly $1$, indicating that sparsemax training has not theoretically guaranteed the suppression of irrelevant tail mass, contradicting the goal of suppressing irrelevant tail mass.
> * **Empirical verification.** To validate our theoretical analysis, we also measure the *realized* softmax tail mass for models trained with Sparsemax. On 100 randomly selected AlpacaEval test examples, we record the average softmax probability of the top-3 tokens, as well as the cumulative mass on non-top-3 tokens and on tokens outside the sparsemax support. It shows the tail probabilities are indeed non-negligible.
>
> | Metric               | $1^{st}$ Prob | $2^{nd}$ Prob | $3^{rd}$ Prob | Non-Top-3 Prob | `Non-Support Prob` |
> | :------------------- | :-------------: | :-------------: | :-------------: | :------------: | :------------------: |
> | **Sparsemax**  |      0.220      |      0.047      |      0.031      |     0.702     |      `0.553`      |
> | **Sparsemax+** |      0.853      |      0.096      |      0.026      |     0.025     |      `0.069`      |
>
> *Table 1: Comparison of softmax tail behaviour between Sparsemax and Sparsemax+.*
>
> We have included the above points in the revised manuscript (**Remark 2,** **Appendix D).**
>
> ---
>
> > **W2:** The proposed loss is simple, compressing the tailed probability to 1 single probability. To support that the proposed one is better, the authors should provide a result like Corollary 5 with a tighter bound (if possible). Otherwise, it is hard to tell whether the proposed loss resolved the raised theoretical issue.
>
> **Response:**
> Thank you for raising this point. We address it on two levels:
>
> * As we have claimed in Concern 1, there exists a tight upper bound under the sparsemax one-hot assumptions:
>   $$
>   \sum_{k>m} p_{(k)}^{\mathrm{sf}}\le \frac{(K-m)e^{-\gamma}}{\Omega_{\min}},
>   $$
>
> where $\Omega_{\min} = 1 + (m-1)e^{-B} + (K-m)e^{-\gamma}$.  The inequality holds given $z_y = 0, z_{(2)} = \cdots = z_{(m)} = -B, z_{(m+1)} = \cdots = z_{(K)} = -\gamma$. And this new upper bound is strictly increasing in $K$ and approaches $1$ as $K \to \infty$.
>
> * **How Sparsemax+ resolves the issue.** Standard sparsemax training may lead to a “non-negligible admissible tail”. In contrast, the Sparsemax$+$ loss adds the penalty:
>
> $$
> L_{\mathrm{suppress}} = -\log(1 - \sum_{i \notin S^{\sf sp}(\mathbf{z}), i \neq y} p_i^{\mathrm{sf}}).
> $$
>
>   This explicitly minimizes the tail mass during training. It forces the model to push tail logits far below the margin ($z_{\text{tail}} \ll -\gamma$), ensuring that the *realized* tail mass is negligible.
>
> * **Empirical Verification.** As shown in Table 1, while the standard Sparsemax model exhibits a realized tail mass of $\mathbf{0.553}$, the Sparsemax$+$ model reduces this to $\mathbf{0.069}$. It demonstrates that the proposed loss effectively resolves the issue in practice by suppressing the realized tail.
>
> We have clarified in the revised manuscript (**Appendix C.10**).

---

> ### Author Response · Authors · 2025-11-22
> **Response to Reviewer 2nRT (continued)**
>
> > **W3:** The definition of sparsemax (lines 154–157) appears to be ill-defined. Specifically, the threshold function is defined in terms of the support set, yet the support set itself is defined based on the threshold function, resulting in an apparent circular definition.
>
> **Response:** Thank you for the insightful comment. You are correct that the expression in the manuscript described a property rather than the primary definition. In the revision, we will adopt the original non-circular definition of sparsemax and clearly separate the definition from the derived properties.
>
> For completeness, the formal definition is:
> Let $\mathbf{z} \in \mathbb{R}^K$ and let $z_{(1)} \ge z_{(2)} \ge \cdots \ge z_{(K)}$ denote the sorted coordinates of $\mathbf{z}$. Define $k(\mathbf{z}) := \max\\big\\{ k \in \{1,\dots,K\} \mid 1 + k z_{(k)} > \sum_{j=1}^{k} z_{(j)} \\big\\}$. The threshold is $\tau(\mathbf{z}) = \frac{\sum_{j=1}^{k(\mathbf{z})} z_{(j)} - 1}{k(\mathbf{z})}$. Then sparsemax is given elementwise by $\textrm{sparsemax}(\mathbf{z})_i = \max\\{z_i - \tau(\mathbf{z}), 0 \\}$. The support set is thus $S^{\mathrm{sp}}(\mathbf{z}) = \\{i : \operatorname{sparsemax}(\mathbf{z})_i > 0\\}$, which is a consequence of the definition rather than a prerequisite.
>
> ---
>
> > **W4:** Lemma 3, $L_{sp}(z,y)$ is not defined before. Leaving readers to guess whether it is referred to “we train with sparsemax and optimize a modified Fenchel–Young loss tailored to this mapping” mentioned in the introduction.
>
> **Response:** We apologize for the ambiguity. In Lemma 3, $\mathcal{L}_{\mathrm{sp}}(\mathbf{z},y)$ refers to the supervised loss that applies the sparsemax probability mapping to the logits (i.e., $\mathbf{p}=\mathrm{sparsemax}(\mathbf{z})$), and can be instantiated as the Fenchel–Young loss in the paper.
>
> In the revision, we rename it to $\mathcal{L}(\mathbf{p},y)$ to avoid any notational ambiguity with other losses used in the paper.  Also, we cite the original paper of sparsemax for reference [1].
>
> [1] Martins, A., \& Astudillo, R. (2016, June). From softmax to sparsemax: A sparse model of attention and multi-label classification. In International conference on machine learning (pp. 1614-1623). PMLR.

---

### Official Review · Reviewer_GaaU · 2025-11-07

**Soundness:** 3
**Presentation:** 3
**Contribution:** 3
**Rating:** 6
**Confidence:** 3

**Summary:**

The authors propose a new normalization method from logits to the probability distribution. By analyzing the behavior of the new normalization method, the authors point out that the method is only good for training but not for inference. Thus, the authors give the overall framework that the model is trained with the new method (i.e., Sparsemax+) and is inferred with softmax. The experimental results show that the framework increases the pass@N a lot when N is large.

**Strengths:**

1. To preserve diversity, the authors introduce a new normalization layer called Sparsemax+.

2. After analyzing the properties of Sparsemax and softmax, the authors choose to use different normalization layers for training and inference.

3. The performance increases a lot when the number of rollout samples is large.

**Weaknesses:**

1. It is unclear what the mismatch of using Sparsemax+ and softmax will cause.  The model performance is not as good as GEM for pass@N when N is small (e.g., N = 2). Is it because of the mismatch?

2. Sparsemax+ does not seem to be tunable. In softmax, there is a temperature parameter that can control the final results. Also, in GEM, there is a hyperparameter that can balance the KL term and the objective.

3. For $i,j \notin S^{sp}$, $\frac{\partial }{\partial u}(p_i - p_j) < 1$, what is different between this result and softmax?

**Questions:**

See Weaknesses.

---

> ### Author Response · Authors · 2025-11-22
> **Response to Reviewer GaaU**
>
> We thank the reviewer for valuable comments. Below is our response to your concerns.
>
> > **W1:** It is unclear what the mismatch of using Sparsemax+ and softmax will cause. The model performance is not as good as GEM for pass@N when N is small (e.g., N = 2). Is it because of the mismatch?
>
> **Response:** Our choice of training with Sparsemax+ and decoding with softmax is *intentional* and directly follows the Tail Suppressed Plausible Diversity (TSPD) objective in Definition 1. *Sparsemax+ is only used to shape the logits during training; at test time we apply softmax to the same logits.* Thus there is no train-test mismatch in the sense of optimizing one objective and sampling from a different set of logits.
>
> **Why small- pass@N is slightly lower.** TS$^2$ intentionally reallocates probability mass from the top-1 token into other plausible tokens, thereby preserving diversity and avoiding mode collapse. This necessarily reduces small-$N$ pass@N (e.g., $N=2$) relative to methods that concentrate almost all mass on a single mode. However, this is the intended trade-off; as illustrated in Fig. 1, TS$^2$ yields a richer set of plausible completions, which is beneficial in open-ended settings where multiple valid answers exist and $N$ is moderately large ($N>2$). Using Sparsemax also at test time would only further sharpen this effect and is therefore not preferred for our diversity-oriented objective.
>
> ---
>
> > **W2:** Sparsemax+ does not seem to be tunable. For softmax, there is a temperature control which would influence the final result and similarly for GEM, there is a hyperparameter that can balance the KL term and objective.
>
> **Response:** We respectfully disagree that Sparsemax+ is not tunable. TS$^2$ exposes clear knobs both at training and inference time.
>
> **1. Training time control via $\alpha$.**
> In Eq. (4), the coefficient $\alpha$ multiplies the out-of-support suppression term and directly controls how aggressively softmax tail mass is penalized. When $\alpha \to 0$, the loss reduces to standard sparsemax: many low-logit tokens retain non-negligible mass and diversity is very high. As $\alpha$ increases, tail tokens are suppressed more strongly and probability concentrates on the top few logits. The table below (also present in **Appendix C.1 Table 3**) shows a smooth diversity–winrate trade-off for different values of $\alpha$, with $\alpha=0.25$ used in our main experiments.
>
> **Effect of $\alpha$ on Chat (AlpacaEval):**
>
> | Setting         | Winrate (%)     | n-gram          | 100-self-BLEU   | Sent-BERT       |
> | :-------------- | :-------------- | :-------------- | :-------------- | :-------------- |
> | $\alpha$=0.25 | **33.12** | **23.78** | **53.87** | **12.80** |
> | $\alpha$=0.50 | 31.44           | 20.54           | 51.34           | 11.61           |
> | $\alpha$=0.75 | 30.84           | 19.07           | 49.29           | 10.96           |
> | $\alpha$=1.00 | 30.39           | 18.13           | 48.05           | 10.30           |
> | $\alpha$=5.00 | 25.34           | 13.44           | 41.42           | 8.12            |
>
> **2. Training time control via logit scale $c$.**
> There is also an internal logit scale $c$ in $L_{\mathrm{spm}^+}(c\\mathbf{z};y)$, which plays a role analogous to temperature. In all reported experiments we simply set $c = 1$, which already yields strong performance.
>
> **3. Inference time control via softmax temperature.**
> At test time, TS$^2$ decodes with standard softmax, so the usual softmax temperature can be tuned exactly as in vanilla SFT. We fix it to 1 in all reported results, but practitioners can further smooth or sharpen the distribution if desired.
>
> Thus, similar to softmax and GEM, TS$^2$ provides explicit, practically useful hyperparameters ($\alpha$, $c$, and softmax temperature) to tune the diversity–accuracy balance.
>
> ---
>
> > **W3:** For $i, j \notin S^{\mathrm{sp}},\ \frac{\partial}{\partial u}\bigl(p_i - p_j\bigr) < 1$, what is different between this result and softmax?
>
> **Response:**
> Sorry for the confusion. The key difference lies in how out-of-support tokens respond to logit perturbations.
>
> **Sparsemax case:** For sparsemax, $i,j \notin S^{\mathrm{sp}}$ implies $p_i^{\mathrm{sp}} = p_j^{\mathrm{sp}} = 0$. Once a token falls outside the support, its probability is clamped to zero and becomes insensitive to further changes in its logits. In other words, sparsemax enforces a hard tail truncation: out-of-support tokens are completely frozen.
>
> **Softmax case:** By contrast, softmax assigns strictly positive mass to all tokens, so even tail tokens remain responsive.
>
> To sum up, we have:
>
> **Sparsemax (sp):**
>
> $$
> \frac{\partial}{\partial u}\bigl(p_i^{\mathrm{sp}} - p_j^{\mathrm{sp}}\bigr) =
> \begin{cases} 0 & i,j \notin \mathcal{S}^{\sf sp}\\\\
> 1 & i,j \in \mathcal{S}^{\sf sp}\end{cases}
> $$
>
> **Softmax (sf):**
>
> $$
> 0 < \frac{\partial}{\partial u}\left(p^{\sf sf}_i-p^{\sf sf}_j\right) < 1, \quad i,j \in \mathcal{S}^{\sf sf}
> $$

---

> > ### Comment · Reviewer_GaaU · 2025-11-27
> >
> > Thanks for the authors' detailed response. I have no more questions, and I will maintain my score.

---

> > > ### Author Response · Authors · 2025-11-27
> > >
> > > Thank you again for your time and valuable comments. We appreciate your positive assessment of our paper.

---

### Author Response · Authors · 2025-11-22
**To all Reviewers**

We thank the reviewers for their thoughtful comments and constructive feedback. We are encouraged that all reviewers acknowledged the significant improvements brought by our method.

We have uploaded a revised version of the paper. Below, we summarize the major updates and additional experiments included in the revision, which address the concerns raised regarding baselines, theoretical clarifications, and qualitative analysis.

* **Additional Baselines **Label Smoothing** and **$\alpha$-Entmax** (Appendix C.2)** Experiments show that while these methods increase diversity, they do so at a significant cost to win-rate. TS$^2$ achieves a superior trade-off, maintaining the highest win-rate (33.12%) while delivering strong diversity.
* **Theoretical Derivation for tightness of the upper bound in Corollary 5 (Remark 2, Appendix D, Appendix C.10 Figure 7).** We expanded the discussion as Remark 2 to clarify the tightness of the softmax tail upper bound, with detailed proof in Appendix D. We also provide empirical support in Figure 7 to demonstrate that our proposed Sparsemax+ loss substantially suppresses the softmax tail mass.
* **Hyperparameter Sensitivity & Ablation (Appendix C.1).** We provided a detailed ablation study on the suppression coefficient $\alpha$ and the logit scale $c$. We demonstrate that performance varies smoothly with $\alpha$, with a robust "sweet spot" around $\alpha \in [0.25, 0.75]$ where the model balances tail suppression and diversity effectively.
* **Multi-Turn & Reasoning Capabilities.** To demonstrate that TS$^2$ preserves reasoning integrity and conversational consistency: (1) **Multi-Turn Benchmark (Appendix C.3):** We evaluated TS$^2$ on **MT-Bench-101**. The method shows consistent improvements across capability groups (Context Memory, Reasoning, etc.) compared to CE and GEM. (2) **Qualitative Reasoning Analysis (Appendix C.9):** We added examples from *GSM8K* (Chain-of-Thought) showing that TS$^2$ generates diverse phrasings while maintaining coherent, correct multi-step reasoning traces. (3)**Open-Ended Generation (Appendix C.9):** We included qualitative comparisons on career advice prompts, illustrating how TS$^2$ produces distinct, non-redundant perspectives compared to the repetitive templates often seen in CE-SFT.
* **Related Work Expansion.** We added a dedicated paragraph in the **Related Work** section discussing the literature on miscalibration and overconfidence in LLMs, explicitly connecting our work to existing calibration techniques and confidence elicitation methods.

We believe these revisions strengthen the paper and well address the reviewers' concerns. We look forward to further discussions.

---

> ### Author Response · Authors · 2025-11-27
>
> Dear Reviewers,
>
> Thank you for your time and valuable comments.
>
> We believe we have addressed all concerns you raised regarding our submission. As the discussion period is ending soon, we would greatly appreciate your response to our rebuttal or any additional feedback.
>
> Best Regards,
> Authors

---

### Author Response · Authors · 2025-12-02
**Rebuttal Summary for the Area Chair**

**Dear Area Chair,**

We understand the complexity caused by the recent review reset. To assist in your assessment, we present this summary demonstrating that our TS$^2$ was already well-received by reviewers for its novelty and empirical strength, and that every concern raised has been addressed in the revised manuscript with new baselines, theoretical proofs, and qualitative analysis.

`GaaU`, `uXcp`, `jH36` all state the soundness, presentation and contribution as ***good*** or above, each giving an overall score of ***6***. All four reviewers acknowledged the significance of the problem and the effectiveness of our solution. The initial feedback highlights that the core contribution is solid:

* `GaaU`: "The performance increases a lot when the number of rollout samples is large"
* `2nRT`: "the results are comprehensive and convincing", "The method outperforms strong baselines such as GEM not only in accuracy but also in diversity metrics"
* `uXcp`: the reviewer highly acknowledge the novelty, quality, clarity and significance of our work
* `jH36`: "The results show strong performance and diversity..., compared against existing SOTA (GEM)"

Below is how we address the concerns raised across the reviews.

* **Hyperparameter sensitivity**[`GaaU`, `uXcp`].  We included this study in **Appendix C.1.**
* **Tightness of the upper bound in Corollary 5** [`2nRT`]. We included the theoretical derivation for tightness of the upper bound in **Remark2 and Appendix D,** which demonstrates that the proposed *Sparsemax+* loss provides a mathematically verifiable improvement over standard *Sparsemax*.
* **Missing baselines**[`uXcp`, `jH36`]**.** We included those established regularization techniques, specifically Label Smoothing or $\alpha$-Entmax suggested by `uXcp`, `jH36` in **Appendix C.2.** Also, we included the discussion for these studies in the **Related Work section.**
* **Whether high diversity negatively impacts factual consistency, long reasoning chains or multi-turn datasets** [`uXcp`, `jH36`]**.** We included empirical results in **Appendix C.3** and **C.9**, showing our TS$^2$ preserves reasoning integrity and conversational consistency.

Thank you for your time in handling our submission.

Best Regards,

Authors

---

### Meta-Review · Area_Chair_DHBE · 2026-01-08

**Summary:**

Paper proposes modification of sparsemax loss to improve/preserve diversity when SFTing retrained models. They add an explicit probability suppression loss for non-top tokens. This works very well to improve output diversity when compared to SOTA methods across multiple datasets provides and two models. Reviewers had concerns about specifics of the algorithm, the strength of theoretical guarantees, missing baselines, and hyperparameter sensitivityWhile the overall paper is written clearly, there were lack of rigor in mathematical definitions and formulations. Reviewers also had subtler questions like performance on specialized tasks which require long reasoning traces and multi-turn datasets, and effect on factual accuracy. AC also notes that authors do not provide any code for verification and the technical novelty might be limited since it is a simple modification to sparsemax.

**Reviewer Concerns:**

During rebuttal authors revised the manuscript for clarity, added new experiments and addressed most of the concerns except those around factual accuracy. From the new experiments we see that the method might be sensitive to the choice of the hyper-parameter alpha.

**Reviewer Scores:**

Assume that Reviewer jH36 will increase their score to 7, and Reviewer 2nRT will increase their score to 5, and all other reviewers will their maintain scores.

---

### Decision · Program_Chairs · 2026-01-26

Accept (Poster)